# DimPO: Dimensionality Reduction for Attention using Preference Optimization

## Abstract

Large Language Models (LLMs) require substantial memory and computation time, particularly for long-context tasks. To handle long sequences, LLMs use KV caches, whose memory size grows linearly with the number of tokens. In this work, we focus on reducing KV cache memory by projecting key and query vectors into learned lower-dimensional spaces. We pose the problem - previously solved with triplet loss for Locality Sensitive Hashing (LSH) - as a preference optimization problem. We show that the preference optimization approach performs better mostly on higher dimensions indicating its potential for training attention in reduced dimensions. To address this, we introduce DimPO, a novel reference-model-free, listwise preference optimization loss. We demonstrate that DimPO more accurately preserves attention distributions in reduced dimensions compared to both existing preference optimization losses and triplet loss. Building on this, we apply DimPO-based dimensionality reduction to the attention layers of LLaMA3-[1B, 3B, 8B], Qwen2.5-7B and Qwen3-4B instruct models. On general benchmark tasks, DimPO Attentions reduces KV cache memory by 10-15% while maintaining 95% of performance. Larger models using DimPO Attentions on long-context tasks also exhibit only a marginal performance drop.

## 1 Introduction

LLMs have demonstrated state-of-the-art performance across a wide range of tasks but require massive computational resources and incur substantial costs not only for training but also for inference (Pope et al., 2023; Zhang et al., 2023). Recently, generative models supporting very long input contexts - up to 128k tokens - have been released (OpenAI et al., 2024; Grattafiori et al., 2024; Yang et al., 2025). However, this introduces additional computational and memory challenges, as every input token must be considered when computing attention for each newly generated query token.

To avoid recomputing the key and value vectors of previously processed tokens, LLMs store them in a KV cache (Pope et al., 2023). However, with each additional input token, the attention computation must consider one more token than before, causing both the computational cost and the memory footprint of the KV cache to grow linearly with the total number of input and generated tokens (Li et al., 2025).

A variety of works have addressed the problem of optimizing the KV cache, focusing mostly on encoding or compressing the set of key and value vectors to reduce the number of stored vectors (Zhang et al., 2023; Tang et al., 2024; Liu et al., 2024; Singhania et al., 2024). In contrast, in this work we compress along a different axis: rather than reducing the number of stored vectors, we reduce the dimensionality of each individual key and query vector. This could have two benefits: (i) key vectors occupy less memory in the KV cache, and (ii) attention weight computation becomes faster because dot products involve shorter vectors. The value vectors remain unmodified, preserving the semantic information they carry as long as the attention weight distribution remains sufficiently close to the original.

Our objective is therefore to train a projection that maps query and key vectors into a low-dimensional space such that the resulting attention weight distribution remains close to that of the original, full-dimensional attention, while introducing minimal computational overhead during the projection itself.

Prior works have also projected keys and queries into lower-dimensional spaces, typically for the purpose of grouping them into buckets to enable sparse or blockwise attention (Wang et al., 2020; Chen et al., 2025; Zeng et al., 2025). LSH-based methods, for instance, use multiple random projections (Kitaev et al., 2020), followed by learnable projections optimized with a triplet loss (Chen et al., 2021b). These approaches generally require multiple hash functions-i.e., multiple projections-because any single projection alone does not preserve enough information about the attention distribution (Kitaev et al., 2020; Chen et al., 2021b; 2025).

Inspired by recent advances in preference optimization for post-training alignment of LLMs (Rafailov et al., 2023; Meng et al., 2024), we frame the problem of dimensionality reduction for attention as a preference optimization problem. Our goal is to learn a linear layer that projects queries to relate more strongly to relevant projected keys and less to irrelevant ones, so that the resulting attention weight distribution matches the behavior of the full-dimensional model as closely as possible.

Preference optimization for post-training alignment of LLMs initially relied on reference models to compute losses (Rafailov et al., 2023; Ethayarajh et al., 2024; Liu et al., 2025). Over time, methods have been developed that do not require a reference model (Hong et al., 2024; Meng et al., 2024). This is particularly relevant for our setting: in the case of attention dimensionality reduction, it is unclear what a reference model would even be. Furthermore, typical preference losses are pairwise, comparing a chosen and a rejected response, which simplifies data collection. Our scenario is different: for each query, we need to capture the full preference ordering across a list of keys. Constructing all pairwise comparisons is theoretically possible but computationally prohibitive and practically infeasible. While listwise preference losses exist (Liu et al., 2025), they typically assume a reference model, which we do not have. These challenges motivate the need for a custom, reference-free loss function tailored to attention dimensionality reduction.

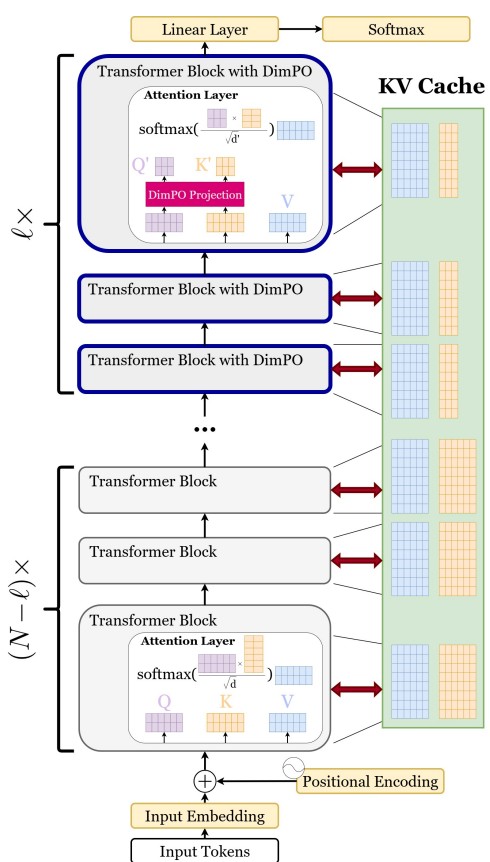

To address this, we introduce DimPO, a novel listwise, reference-free preference optimization loss specifically designed for dimensionality reduction of query and key vectors. DimPO captures the full preference ordering without relying on a reference model and, as we show, significantly outperforms existing approaches, including pairwise preference optimization losses, triplet loss, and random projection baselines, indicating its potential for use in real-world model deployments and practical tasks.

Figure 1: Decoder model architecture with integrated DimPO-based attention projection applied to last $\ell$ attention layers.

To validate the effectiveness of our KV cache reduction approach, we conduct experiments on several instruction-tuned models, including LLama3.2-[1B,3B]-Instruct, Llama3.1-8B-Instruct (Grattafiori et al., 2024), Qwen3-4B-Instruct (Yang et al., 2025), and Qwen-2.5-7B-Instruct (Yang et al., 2024). We apply our DimPO-trained projections progressively from the top attention layers downward, as illustrated in Figure 1. This top-down approach ensures that errors from modified layers have a limited impact on subsequent attention layers.

We measure how many attention layers can be modified using DimPO-based projections and what proportion of KV cache memory can be saved without causing a significant drop in model performance. Evaluations are performed on both general benchmarks and long-context tasks. Our results indicate that even a 10% reduction in KV cache memory leads to only a marginal performance drop

on general tasks. For long-context tasks, larger models maintain performance similar to general tasks at around 10% memory savings, while smaller models experience a more noticeable decline.

In Sections 3.1 we formalize the attention dimensionality reduction problem and in Section 3.2 we introduce the DimPO loss function. In Section 3.4, we compare our method with other preference optimization techniques and baseline approaches in dimensionality reduction of attention. Section 4 presents experiments evaluating DimPO-based projections on real tasks. We investigate how many attention layers can be modified and quantify the corresponding KV cache memory savings, while ensuring that model performance remains largely unchanged.

## 2 RELATED WORKS

**KV Cache Optimization** The KV cache memory bottleneck in long-context LLMs constrains batch size and maximum prompt length, motivating strategies to reduce key and value vectors while maintaining accuracy. H2O (Zhang et al., 2023), SnapKV (Li et al., 2024), and Keyformer (Adnan et al., 2024) use heuristics during prefilling to select tokens for decoding. Quest (Tang et al., 2024) and Loki (Singhania et al., 2024) apply dynamic sparsity during inference to reduce KV cache loading without eviction. KIVI (Liu et al., 2024) and QServe (Lin et al., 2025) reduce KV cache via quantization.

**Attention Approximation via Projections** Transformer variants leverage projections to approximate or accelerate attention. Sparse attention methods (BigBird (Zaheer et al., 2020), Longformer (Beltagy et al., 2020), SparseAxial (Ho et al., 2020)) compute selected blocks or local windows. LSH-based approaches (Reformer (Kitaev et al., 2020), KDEformer (Zandieh et al., 2023), ScatterBrain (Chen et al., 2021a), MagicPIG (Chen et al., 2025)) approximate attention via locality-sensitive hashing. Mongoose (Chen et al., 2021b) builds on LSH with learnable projections trained via triplet loss to group semantically similar keys and queries. Low-rank and linear attention methods (Linformer (Wang et al., 2020), Performer (Choromanski et al., 2021), Nyströmformer (Xiong et al., 2021)) project the attention matrix to lower-dimensional spaces. Top-$k$ mechanisms (Unlimiformer (Bertsch et al., 2023), IceFormer (Mao et al., 2024), ZETA (Zeng et al., 2025)) use projections and dimension reduction for efficient token selection.

**Preference Optimization** Preference optimization aligns models with desired outputs using ranked feedback, often via *chosen/rejected* pairs. DPO (Rafailov et al., 2023) simplifies RLHF (Christiano et al., 2017) by removing the reward model and framing alignment as a single-stage classification, still using a reference model to prevent distributional drift. Variants include ORPO (Hong et al., 2024) (odds-ratio), SimPO (Meng et al., 2024) (average log-probability as implicit reward), CPO (Xu et al., 2024) (contrastive learning for machine translation), KTO (Ethayarajh et al., 2024) (prospect-theory utility for binary labels, still needing a reference), and listwise objectives like LiPO (Liu et al., 2025), considering multiple ranked responses while relying on a reference model.

## 3 LEARNING LOW-DIMENSIONAL ATTENTION PROJECTION

In this section, we formalize the problem of dimensionality reduction for key and query vectors, aiming to minimize the KL divergence between the original attention weight distribution and the distribution computed from the reduced projected vectors. Although more complex non-linear projection approaches could be considered, we require minimal runtime overhead during inference in generative language models. For this reason, we focus on learnable linear layer projections and evaluate several loss functions for training such a projection layer within a Siamese framework (Pai et al., 2019). To this end, we introduce our novel loss function, DimPO, and compare it with other preference optimization losses as well as baseline projection approaches.

### 3.1 PROBLEM FORMULATION

Let $l$ denote a transformer attention layer with query vectors $Q_l \in \mathbb{R}^{N \times d}$ and key vectors $K_{l,q} \in \mathbb{R}^{m \times d}$ for a given $q \in Q_l$ where $N$ is the total number of queries, $m$ is the number of keys per each query and $d$ is the original embedding dimension. Our goal is to learn a shared linear projection

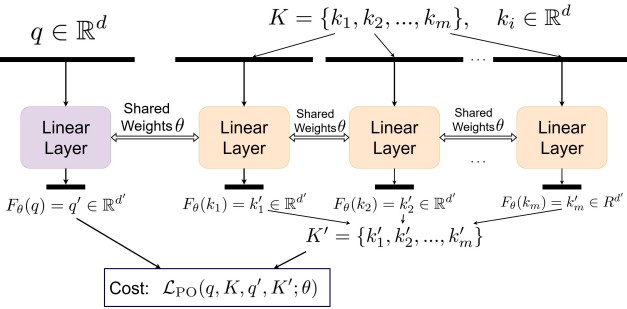

Figure 2: Siamese network architecture for query $q$ and its associated list of keys $K$ using preference optimization loss.

function

$$F : \mathbb{R}^d \to \mathbb{R}^{d'},$$

with $d' \ll d$, that maps both queries and keys into a lower-dimensional space such that the resulting attention weight distribution is as close as possible to the original one. Formally, we solve

$$F^* = \arg\min_F \frac{1}{N} \sum_{q \in Q_l} \mathrm{KL}\left( \underbrace{\mathrm{Softmax}\left( \frac{q K_{l,q}^\top}{\sqrt{d}} \right)}_{\text{original attention}}, \underbrace{\mathrm{Softmax}\left( \frac{F(q) F(K_{k,q})^\top}{\sqrt{d'}} \right)}_{\text{projected attention}} \right),$$

where $\mathrm{KL}(\cdot, \cdot)$ denotes the Kullback–Leibler divergence between the original and projected attention distributions.

To minimize this divergence in a computationally efficient way, we restrict $F$ to a single linear projection layer, $F(x) = Wx$ with $W \in \mathbb{R}^{d' \times d}$, ensuring negligible inference overhead in generative models. We then train $W$ using a Siamese framework, as illustrated in Figure 2, where each query is paired with its ranked keys (in the case of pairwise losses, only two ranked keys are used) according to their importance for the given query, derived from the original attention weight distribution. This setup is formulated as a preference optimization problem: the projection $F$ is optimized such that keys with higher original attention weights have higher similarity to the query representation (in terms of dot product), while less relevant keys have lower similarity.

The probability assigned by the linear layer parameters $\theta$ to a given key $k$ (either from a key pair or the full list $K_{l,q}$ associated with the current training instance) after projection $F(k) = k'$ and the corresponding query $F(q) = q'$ is computed as

$$\pi_\theta(k' \mid q') = \mathrm{Softmax}\left( \frac{q' K_{l,q}'^\top}{\sqrt{d'}} \right),$$

where $K_{l,q}' = F(K_{l,q})$ denotes the set of projected keys and $k' \in K_{l,q}', k \in K_{l,q}$. For pairwise losses, this probability computation is limited to the two selected keys in $K_{l,q}$, whereas for listwise loss functions, all keys associated with the given query are used.

We evaluate the quality of the learned projection $F$ using two complementary metrics, averaged over all $N$ evaluation instances $(q, K_{l,q})$ for all $q \in Q_l$, where $V_{l,q} \in \mathbb{R}^{N \times d}$ denotes the value vectors for layer $l$ associated with query $q$.

1. **Attention weights KL Divergence:** the average Kullback–Leibler divergence between the original attention distribution and the projected one for each query-key set:

$$\mathrm{KL} = \frac{1}{N} \sum_{q \in Q_l} \mathrm{KL}\left( \mathrm{Softmax}\left( \frac{q K_{l,q}^\top}{\sqrt{d}} \right), \mathrm{Softmax}\left( \frac{F(q) F(K_{l,q})^\top}{\sqrt{d'}} \right) \right),$$

2. **Attention output MSE:** the average mean squared error between the original attention output and the output after projection for each query-key set:

$$\mathrm{MSE} = \frac{1}{N} \sum_{q \in Q_l} \left\| \mathrm{Softmax}\left( \frac{q K_{l,q}^\top}{\sqrt{d}} \right) V_{l,q} - \mathrm{Softmax}\left( \frac{F(q) F(K_{l,q})^\top}{\sqrt{d'}} \right) V_{l,q} \right\|_2^2.$$

## 3.2 DIMPO: DIMENSIONALITY-REDUCED PREFERENCE OPTIMIZATION

Preference optimization research has predominantly focused on pairwise comparisons (Rafailov et al., 2023; Hong et al., 2024; Xu et al., 2024; Meng et al., 2024). While effective for many tasks, this is not practical for dimensionality reduction of query-key interactions, where each query is compared against many keys. Using a pairwise loss, one can either select a single positive-negative key pair for each query, losing information about relationships with other keys, or consider all possible key pairs, creating $m(m-1)/2$ training instances for each query for $m$ context tokens, which dramatically increases computational and memory costs making the training infeasible to complete. For this reason, it is more practical in this setting to adopt listwise preference optimization losses. One of the most popular pairwise preference optimization methods is DPO (Rafailov et al., 2023):

$$\mathcal{L}_{\text{DPO}}(\pi_\theta; \pi_{\text{ref}}) = -\mathbb{E}_{(x,y_w,y_l)\sim\mathcal{D}}\Big[\log\sigma\big(\beta\log\frac{\pi_\theta(y_w\mid x)}{\pi_{\text{ref}}(y_w\mid x)} - \beta\log\frac{\pi_\theta(y_l\mid x)}{\pi_{\text{ref}}(y_l\mid x)}\big)\Big],$$

which fits naturally into the Bradley-Terry (BT) (Bradley & Terry, 1952) ranking model:

$$p(y_w \succ y_l \mid x) = \sigma\big(r(x,y_w) - r(x,y_l)\big),$$

where $y_w$ denotes the preferred response, $y_l$ the non-preferred response, and $r(x,y)$ is the reward function. LiPO (Liu et al., 2025) generalizes this BT model to a list of responses $y = (y_1,\ldots,y_K)$:

$$p(y_1 \succ y_2 \succ \cdots \succ y_K \mid x) = \prod_{i=1}^{K}\frac{\exp(s_i)}{\sum_{j=i}^{K}\exp(s_j)},$$

where $s_i = r(x,y_i)$ denotes the score of response $y_i$. This reduces exactly to the pairwise BT model when $K = 2$. In the formulation of listwise loss LiPO, the training dataset consists of lists of responses with corresponding real-valued labels $\psi = (\psi_1,\ldots,\psi_K)$, and a ranking loss is applied over all pairs within the list:

$$\mathcal{L}_{\lambda\text{-loss}}(\pi_\theta) = -\mathbb{E}_{(x,y,\psi)\sim\mathcal{D}}\left[\sum_{\psi_i>\psi_j}\Delta_{i,j}\log\big(1 + e^{-(s_i-s_j)}\big)\right],$$

where

$$\Delta_{i,j} = \left|\frac{1}{D(\tau(i))} - \frac{1}{D(\tau(j))}\right|, \quad G_i = 2^{\psi_i} - 1, \quad D(\tau(s_i)) = \log(1 + \tau(s_i)).$$

Here, $\tau(s_i)$ denotes the rank position of $y_i$ in the permutation induced by the scores $s$, and the scores are defined as

$$s_i = \beta\log\frac{\pi_\theta(y_i\mid x)}{\pi_{\text{ref}}(y_i\mid x)},$$

with $\beta > 0$ controlling the sharpness of the preference optimization.

In our setting, the reference model $\pi_{\text{ref}}$ is not available. Furthermore, SimPO argues that using a reference model during training is inconsistent with inference, in which no reference is present, which can generate inaccurate responses (Meng et al., 2024). However, in our setting, we treat $\pi_{\text{ref}}$ as a uniform distribution and approximate it with a constant $1/c$, which cancels in the log-ratio in the $e^{-(s_i-s_j)}$ term of the sum in the LiPO loss equation

$$s_i - s_j = \beta\log\frac{\pi_\theta(y_i\mid x)}{\pi_{\text{ref}}(y_i\mid x)} - \beta\log\frac{\pi_\theta(y_j\mid x)}{\pi_{\text{ref}}(y_j\mid x)}$$

$$= \beta\Big(\log\pi_\theta(y_i\mid x) - \log\pi_{\text{ref}}(y_i\mid x) - \log\pi_\theta(y_j\mid x) + \log\pi_{\text{ref}}(y_j\mid x)\Big)$$

$$= \beta\Big(\log\pi_\theta(y_i\mid x) - \log\pi_\theta(y_j\mid x)\Big),$$

yielding

$$s_i = \beta\log\pi_\theta(y_i\mid x).$$

Finally, following SimPO's BT adaptation, which introduces a target reward margin $\gamma > 0$ to ensure that the score difference between better and worse responses is at least $\gamma$ (a margin known to improve

generalization capabilities of classifiers (Boser et al., 1992; Cortes & Vapnik, 1995; Agresti, 2002; Turner & Firth, 2012) ) the pairwise margin-adjusted BT model is defined as

$$p(y_w \succ y_l \mid x) = \sigma\big(r(x, y_w) - r(x, y_l) - \gamma\big).$$

Building on this, we now introduce our preference optimization loss, DimPO:

$$\mathcal{L}_{DimPO}(\pi_\theta) = -\mathbb{E}_{(x,y,\psi)\sim\mathcal{D}}\left[\sum_{\psi_i > \psi_j} \Delta_{i,j} \log\big(1 + e^{-(s_i - s_j - \gamma)}\big)\right], \quad (1)$$

with $\Delta_{i,j}, G_i, D(\tau(s_i)), \tau(s_i)$ and scores $s_i = \beta \log \pi_\theta(y_i|x)$ as defined above. This formulation provides a reference-model-free, listwise, margin-aware preference optimization objective, capturing all key-query interactions efficiently while maintaining theoretical consistency with DPO when $K = 2$.

### 3.3 TRAINING SETUP

Having defined the objective, we now describe the practical training procedure used to obtain the projection $F$. The projection is trained independently for each transformer layer $l$, but a single $W$ is shared across all attention heads within that layer to reduce parameter count and training complexity.

For training, we take the first 4096 tokens from 10 chapters of the training split of the BOOKSUM dataset (Kryscinski et al., 2022), ensuring that each chapter contains at least 4096 tokens. Each chapter thus provides a set of queries paired with lists of 4096 keys. To balance coverage and computational efficiency, we subsample every $H_{l,q}$-th query from each attention head, where $H_{l,q}$ is the number of query heads in layer $l$, resulting in a total of 40,960 training instances per layer. The parameters of the projection function $F$ are then optimized using the Adam optimizer.

For validation, we select 10 chapters from the BOOKSUM validation split, each containing at least 4096 tokens, and use the first 4096 tokens from each chapter. For evaluation, we select 10 chapters from the BOOKSUM test split in the same manner, using the first 4096 tokens of each chapter. Unlike training and validation, during evaluation we include all queries from all attention heads to obtain a complete measure of attention distribution preservation.

Before training, we first derive preference rankings from the key and query vectors. For DimPO, we additionally assign a score to each key by computing the original attention weights $s_i = \text{Softmax}\left(\frac{qK_{l,q}}{\sqrt{d}}\right)_i$. Sorting the keys in descending order by $s_i$ yields a preference ranking.

For DimPO training, we use all keys of each training instance with their exact scores as attention weights $s_i$ and the derived preference ranking. For other methods that use pairwise losses, we select a single key pair per training instance, choosing the highest-ranked key as *chosen* and the lowest-ranked key as *rejected*, following the setup used by Mongoose for triplet-loss training (Chen et al., 2021b). This ensures that all methods (ours as well as other preference optimization losses) are trained on an equal number of training instances (see Appendix C for evidence that, even if more complex training with multiple pairs were used, pairwise methods often stagnate or degrade rather than improve). Detailed hyperparameters used for training each method are provided in Appendix A.

### 3.4 BASELINE COMPARISON

In this section, we compare DimPO against several pairwise, reference-model-free preference optimization methods, namely CPO, ORPO, and SimPO, on the task of attention dimensionality reduction. We also include random projection and triplet loss as baselines, and add PCA projection as a reference. While PCA is neither designed for online training nor aimed at preserving attention-relevant dimensions, it provides a useful comparison, even though it discards variance from low-variance dimensions that may still be crucial for attention.

For evaluation, we train a separate linear projection layer for each attention layer of Llama3.1-8B-Instruct. Results for Qwen and other Llama models are provided in Appendix B. Each projection maps from the original head dimension $d = 128$ to a target dimension $d' \ll 128$ (note that Llama3.2-1B uses $d = 64$, while all other reported models have $d = 128$). Table 1 summarizes the results, averaged across all layers, for $d' \in \{64, 32, 16, 8, 4, 2, 1\}$.

|  | 64 | | 32 | | 16 | | 8 | | 4 | | 2 | | 1 | |
|---|---|---|---|---|---|---|---|---|---|---|---|---|---|---|
|  | KL | MSE | KL | MSE | KL | MSE | KL | MSE | KL | MSE | KL | MSE | KL | MSE |
| Rand | 12.56 | 0.084 | 14.50 | 0.104 | 15.78 | 0.119 | 16.52 | 0.129 | 16.60 | 0.128 | 16.82 | 0.132 | 16.70 | 0.126 |
| PCA | 2.65 | 0.011 | 5.71 | 0.013 | 7.28 | 0.013 | 7.57 | 0.013 | 6.66 | 0.016 | 6.44 | 0.019 | 6.49 | 0.021 |
| Triplet | 3.19 | 0.014 | 3.61 | 0.014 | 3.97 | 0.015 | 4.14 | 0.015 | 4.16 | 0.015 | 4.18 | 0.015 | 4.33 | 0.015 |
| CPO | 2.22 | 0.010 | 1.80 | 0.008 | 2.32 | 0.012 | 3.27 | 0.014 | 3.81 | 0.015 | 4.13 | 0.015 | 4.44 | 0.015 |
| SimPO | 2.90 | 0.010 | 2.11 | 0.009 | 1.94 | 0.009 | 2.17 | 0.009 | 2.58 | 0.012 | 3.05 | 0.016 | 3.86 | 0.028 |
| ORPO | 2.72 | 0.010 | 1.93 | 0.008 | 1.75 | 0.008 | 1.96 | 0.009 | 2.31 | 0.011 | 2.68 | 0.015 | 3.26 | 0.022 |
| **DimPO** | **0.67** | **0.005** | **0.97** | **0.006** | **1.29** | **0.007** | **1.57** | **0.008** | **1.83** | **0.009** | **2.10** | **0.011** | **2.62** | **0.014** |

Table 1: Comparison of different projection approaches for Llama3.1-8B-Instruct. Reported values are attention weights KL divergence and attention output MSE (lower is better), averaged across all attention layers, for different target dimensions $d' \in \{64, 32, 16, 8, 4, 2, 1\}$.

| $d'$ / $\ell$ | 0 | 2 | 4 | 8 | 12 | 16 | 20 | 24 | 28 | 30 | 32 |
|---|---|---|---|---|---|---|---|---|---|---|---|
| 64 | 66 / 0% | 66 / 2% | 66 / 3% | 66 / 6% | 65 / 9% | 64 / 13% | 61 / 16% | 57 / 19% | 52 / 22% | 48 / 23% | 43 / 25% |
| 16 | 66 / 0% | 65 / 3% | 65 / 5% | 64 / 11% | 62 / 16% | 52 / 22% | 42 / 27% | 37 / 33% | 35 / 38% | 35 / 41% | 35 / 44% |
| 4 | 66 / 0% | 65 / 3% | 64 / 6% | 62 / 12% | 54 / 18% | 42 / 24% | 36 / 30% | 35 / 36% | 34 / 42% | 35 / 45% | 35 / 48% |
| 1 | 66 / 0% | 65 / 3% | 64 / 6% | 59 / 12% | 52 / 19% | 39 / 25% | 35 / 31% | 35 / 37% | 35 / 43% | 35 / 47% | 36 / 50% |

Table 2: Average performance across Arc-Challenge, HellaSwag, TruthfulQA-mc2, MMLU and WinoGrande tasks for Llama3.1-8B-Instruct. Each cell reports the average score and the percentage of KV cache memory saved for the number of projected attention layers $\ell$ and target dimension $d'$.

Table 1 shows that learned linear projections outperform random projections, commonly used in LSH-based approaches. Preference optimization losses consistently exceed the Mongoose triplet-loss approach, especially at higher dimensions. Among these, DimPO achieves roughly three times lower error than triplet loss in high-dimensional settings and surpasses other preference optimization losses across all tested $d'$. Overall, DimPO consistently ranks best across all target dimensions, highlighting its potential for training attention in reduced dimensions beyond LSH use cases.

If the learned projection accurately estimates the attention weight distribution in a low-dimensional key-query space, it can enable smaller KV caches and faster inference without sacrificing performance. However, for practical deployment, an open question remains: how far can we safely reduce dimensionality, and in how many layers, before accumulated deviation from the original attention distribution begins to degrade model performance? Even a small perturbation in the first layer could propagate and amplify through the network, whereas a similar perturbation in the final layer may have minimal downstream effect.

## 4 EXPERIMENTS

From the previous section, we know that DimPO outperforms all other methods in estimating attention weight distributions. Therefore, under the same training settings, we train a linear layer with the DimPO loss for each attention layer of the generative model independently. It remains unclear how many attention layers can be safely integrated with this projection without causing a noticeable drop in performance. Since errors in the $i$-th attention layer affect all subsequent layers, while preceding layers remain unaffected, it is sensible to integrate projections starting from the last layer. Noise in the last attention layer does not propagate further, whereas noise in the first layer impacts all following layers. Accordingly, in the experiments that follow, we apply DimPO-based projections starting from the back and progressively extend them toward the beginning, studying the trade-off between performance drop and KV cache memory reduction, as illustrated in Figure 1.

Storing keys and values in the KV cache at their original sizes and head dimensions across all layers is considered 0% KV cache reduction. Conversely, if key vectors were not stored at all, the remaining value vectors (unchanged by our method) would account for 50% of the total KV cache, representing a practical upper bound on achievable reduction. Projecting key vectors to fewer dimensions across more layers reduces the KV cache memory footprint, the primary objective of our method.

| $d'$ / $\ell$ | 0 | 2 | 4 | 6 | 9 | 12 | 15 | 21 | 27 | 32 | 36 |
|---|---|---|---|---|---|---|---|---|---|---|---|
| **64** | 66 / 0% | 66 / 1% | 66 / 3% | 66 / 4% | 65 / 6% | 65 / 8% | 63 / 10% | 61 / 15% | 56 / 19% | 54 / 22% | 52 / 25% |
| **16** | 66 / 0% | 65 / 2% | 64 / 5% | 63 / 7% | 62 / 11% | 57 / 15% | 49 / 18% | 42 / 26% | 39 / 33% | 36 / 39% | 34 / 44% |
| **4** | 66 / 0% | 64 / 3% | 63 / 5% | 62 / 8% | 60 / 12% | 49 / 16% | 43 / 20% | 38 / 28% | 36 / 36% | 35 / 43% | 35 / 48% |
| **1** | 66 / 0% | 64 / 3% | 63 / 6% | 61 / 8% | 59 / 12% | 47 / 17% | 42 / 21% | 38 / 29% | 35 / 37% | 35 / 44% | 35 / 50% |

Table 3: Average performance across Arc-Challenge, HellaSwag, TruthfulQA-mc2, MMLU and WinoGrande tasks for Qwen3-4B-Instruct. Each cell reports the average score and the percentage of KV cache memory saved for the number of projected attention layers $\ell$ and target dimension $d'$.

To evaluate model performance for varying numbers $\ell$ of DimPO-based projected attention layers and projected dimensions $d' \in \{64, 16, 4, 1\}$ from the original 128, we measure five generic benchmark tasks: `Arc-Challenge` (acc_norm) (Clark et al., 2018), `HellaSwag` (acc_norm) (Zellers et al., 2019), `MMLU` (acc) (Hendrycks et al., 2021), `TruthfulQA-mc2` (acc) (Lin et al., 2022), and `WinoGrande` (acc) (Sakaguchi et al., 2021) using harness 0.4.9.1 (Sutawika et al., 2025) in zero-shot settings. The average scores along with the corresponding KV cache memory reduction percentages for different $d'$ and $\ell$ settings are reported in Table 2 for Llama3.1-8B-Instruct and Table 3 for Qwen3-4B-Instruct.

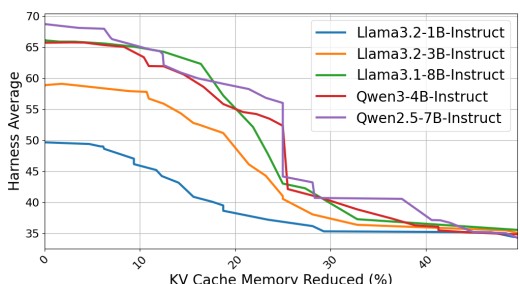

Figure 3: Average performance across Arc-Challenge, HellaSwag, TruthfulQA-mc2, MMLU, and WinoGrande tasks, shown as a function of KV cache reduction.

The tables show the average harness score and the percentage of KV cache memory reduced (see Appendix D for detailed results on all tasks and models, including Llama3.2-1B-, 3B-, 8B-Instruct, Qwen3-4B-, and Qwen2.5-7B-Instruct). Projecting only the last few layers minimally affects performance, even at dimensions as low as 4 or 1, while projecting layers further from the output gradually reduces it. Figure 3 illustrates this trend across all five models, showing that a 10–15% KV cache reduction preserves roughly 95% of the original performance.

The challenges of computational time and large KV cache memory primarily arise in long-context tasks. Based on Figure 3, we selected settings for evaluating the efficiency of DimPO-based projection method on long-context tasks, reducing the KV cache by approximately 6%, 10%, and 12%. We compare these settings across all models (excluding Qwen2.5, whose pretraining did not extensively target long-context tasks) against their base models on all RULER subtasks (Hsieh et al., 2024) available in harness 0.4.9.1 for 4k and 8k context lengths. Table 4 reports averages across all subtasks for the specified context lengths, including throughput in tokens/s.[1]

Unlike generic tasks, smaller models are more sensitive to DimPO-based projections on long-context tasks, particularly Llama 1B, which quickly experienced substantial performance degradation. Larger models, such as Llama3.2-8B-Instruct and Qwen3-4B-Instruct, are more robust and exhibit trends in preserving performance under KV cache reductions similar to those observed in generic tasks, suggesting that the performance impact could be even smaller for larger and more resilient models. In addition to reducing KV cache memory usage, integrating DimPO-based projections into the attention layers also tends to increase token throughput.

Building on these observations, we explore the interaction between DimPO-based projections and the MagicPIG framework, which efficiently optimizes KV cache and attention computation for long-context tasks using LSH-based random projections and CPU-GPU co-design. We investigate whether DimPO can complement MagicPIG by further reducing KV cache usage while maintaining performance. Since MagicPIG currently supports only LLaMA3.1-8B-Instruct among our models, Table 5 compares the base model, its MagicPIG variant, and the MagicPIG variant with DimPO applied to the last 12 layers, corresponding to a 10% reduction of the original MagicPIG KV cache.

---

[1]Throughput was measured using the eager attention implementation for a fair comparison, since other optimized attention implementations either do not support varying key, value, and query dimensions or are not optimized for such cases, which is why we restrict our evaluation to 4k and 8k context lengths.

|  | Average | | Tokens/s | |
|---|---|---|---|---|
|  | **4K** | **8K** | **4K** | **8K** |
| Llama3.2-1B-Instruct (full) | 79.35 | 72.94 | 29.09 | 10.58 |
| **6.25%** $d' = 32, \ell = 4$ | 54.82 | 35.23 | 34.59 | 17.08 |
| **9.38%** $d' = 32, \ell = 6$ | 17.72 | 8.95 | 32.98 | 15.63 |
| **12.50%** $d' = 32, \ell = 8$ | 3.53 | 1.87 | 34.77 | 16.54 |
| Llama3.2-3B-Instruct (full) | 92.56 | 87.31 | 18.94 | 7.16 |
| **6.25%** $d' = 64, \ell = 7$ | 86.04 | 75.68 | 19.36 | 7.91 |
| **10.71%** $d' = 64, \ell = 12$ | 81.83 | 71.14 | 19.32 | 8.39 |
| **12.50%** $d' = 64, \ell = 14$ | 72.61 | 61.84 | 17.65 | 7.81 |
| Llama3.1-8B-Instruct (full) | 95.05 | 93.94 | 13.35 | 4.57 |
| **6.25%** $d' = 64, \ell = 8$ | 94.22 | 91.06 | 13.27 | 4.65 |
| **9.38%** $d' = 64, \ell = 12$ | 93.59 | 89.15 | 13.39 | 4.79 |
| **12.50%** $d' = 64, \ell = 16$ | 90.46 | 85.09 | 20.17 | 10.88 |
| Qwen3-4B-Instruct (full) | 93.86 | 93.08 | 13.37 | 4.98 |
| **6.25%** $d' = 64, \ell = 9$ | 93.28 | 90.19 | 10.94 | 5.99 |
| **10.42%** $d' = 64, \ell = 15$ | 92.01 | 83.92 | 18.10 | 9.53 |
| **12.50%** $d' = 64, \ell = 18$ | 84.84 | 70.27 | 17.92 | 7.07 |

Table 4: Average accuracy and token throughput on RULER long-context tasks for different models in different DimPO projection settings.

|  | LongBench | RULER | | | | |
|---|---|---|---|---|---|---|
|  |  | **4K** | **8K** | **16K** | **32K** | **65K** |
| Llama3.1-8B-Instruct | 37.83 | 95.05 | 93.94 | 93.39 | 87.76 | 84.75 |
| MagicPIG | 35.84 | 92.63 | 92.35 | 91.64 | 86.71 | 83.67 |
| MagicPIG **9.38%** ($d' = 64, \ell = 12$) | 32.58 | 87.59 | 83.41 | 79.56 | 75.29 | 63.66 |

Table 5: Comparison of Llama3.1-8B-Instruct, MagicPIG ($K = 8$, $L = 75$) built on Llama3.1-8B-Instruct, and MagicPIG extended with DimPO-based projections on $d' = 64, \ell = 12$ attention layers, which reduce KV cache memory by 9.38%.

Performance is evaluated on LongBench (Bai et al., 2024) and RULER (Hsieh et al., 2024) tasks across different context lengths, averaging scores over all available subtasks with harness 0.4.9.1 (Sutawika et al., 2025). Despite the additional error introduced by projecting multiple layers, performance decreases gradually and remains reasonably high, highlighting the potential of combining these two approaches to optimize inference for long-context generative tasks.

## 5 CONCLUSION

In this work, we approached attention dimensionality reduction as a preference optimization problem with the goal of reducing KV cache memory. We introduced DimPO, a listwise preference-optimization loss, which consistently outperforms not only existing approaches for projecting key and query vectors but also other reference-model-free preference optimization losses. These projections enable more efficient inference, achieving a 10-15% reduction in KV cache memory with only about a 5% performance drop on generic tasks. While long-context tasks present a greater challenge (particularly for smaller models), larger models maintain performance close to their baselines even when reducing KV cache memory by 10%, indicating that the approach scales well to larger architectures. Beyond introducing a novel preference-optimization loss function and reframing dimensionality reduction as a preference-optimization problem, our work proposes a promising direction for future research by using dimensionality reduction of key and query vectors to optimize KV cache memory usage and attention computation efficiency, addressing two open and critical challenges for scaling large language models. [2]

---

[2]DimPO code is available at: anonymized

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

# A HYPERPARAMETERS OF DIMENSIONALITY REDUCTION APPROACHES

**Hyperparameters.** We report here the hyperparameters used for preference optimization based dimensionality reduction. All methods were trained using the Adam optimizer, with the learning rate specified in the tables. Hyperparameters were tuned on a validation set extracted from BOOKSUM to select the final configuration. To provide a sense of computational cost, we also report the approximate training time per attention layer: DimPO is slower ( 5 min/layer) due to additional attention computations for computing model likelihoods, whereas SimPO, Triplet, ORPO, and CPO are substantially faster ( 10 s/layer). Table 6 lists the final selected settings. Table 7 enumerates all tested values for each hyperparameter across methods, wherever applicable.

| Method | $\beta$ | $\gamma$ | Learning rate | Batch size | Time/layer |
|---|---|---|---|---|---|
| DimPO | 1.0 | 0.0001 | 0.0001 | 1 | ∼5 min |
| SimPO | 1.0 | 1.0 | 0.001 | 32 | ∼10 s |
| Triplet | – | – | 0.0001 | 32 | ∼10 s |
| ORPO | 0.1 | – | 0.001 | 32 | ∼10 s |
| CPO | 1.0 | 0.1 | 0.0001 | 32 | ∼10 s |

Table 6: Final hyperparameter settings for all preference optimization methods.

| Hyperparameter | Tested values |
|---|---|
| $\beta$ | 0.0001, 0.001, 0.01, 0.1, 1.0, 2, 2.5, 5.0 |
| $\gamma$ | 0, 0.00001, 0.0001, 0.001, 0.01, 0.1, 1.0 |
| Learning rate | 1e-5, 1e-4, 1e-3, 1e-2, 0.1 |
| Batch size | 1, 2, 4, 8, 16, 32, 64 |

Table 7: Hyperparameter values explored during tuning for all methods, where applicable.

# B  PERFORMANCE EVALUATION OF PROJECTION APPROACHES

|  | 32 | | 16 | | 8 | | 4 | | 2 | | 1 | |
|---|---|---|---|---|---|---|---|---|---|---|---|---|
|  | KL | MSE | KL | MSE | KL | MSE | KL | MSE | KL | MSE | KL | MSE |
| Rand | 15.02 | 0.087 | 15.92 | 0.092 | 16.37 | 0.095 | 16.49 | 0.103 | 16.62 | 0.099 | 16.42 | 0.102 |
| PCA | 3.83 | 0.014 | 7.09 | 0.016 | 8.04 | 0.017 | 8.06 | 0.020 | 8.46 | 0.026 | 8.13 | 0.026 |
| Triplet | 3.15 | 0.010 | 3.59 | 0.010 | 3.92 | 0.011 | 4.07 | 0.011 | 4.14 | 0.011 | 4.21 | 0.011 |
| CPO | 1.92 | 0.008 | 2.69 | 0.010 | 3.39 | 0.011 | 3.89 | 0.011 | 4.20 | 0.011 | 4.40 | 0.011 |
| SimPO | 1.67 | 0.008 | 1.93 | 0.008 | 2.52 | 0.010 | 3.08 | 0.012 | 3.61 | 0.016 | 4.35 | 0.020 |
| ORPO | 1.62 | 0.008 | 1.86 | 0.008 | 2.38 | 0.009 | 2.78 | 0.011 | 3.02 | 0.012 | 3.69 | 0.016 |
| DimPO | **0.94** | **0.007** | **1.45** | **0.008** | **1.95** | **0.009** | **2.38** | **0.009** | **2.64** | **0.010** | **3.19** | **0.014** |

Table 8: Comparison of different projection approaches for Llama3.2-1B-Instruct. We report the KL Divergence of attention weights and the MSE of attention outputs, averaged over all attention layers, for different target dimensions $d' \in \{32, 16, 8, 4, 2, 1\}$.

|  | 64 | | 32 | | 16 | | 8 | | 4 | | 2 | | 1 | |
|---|---|---|---|---|---|---|---|---|---|---|---|---|---|---|
|  | KL | MSE | KL | MSE | KL | MSE | KL | MSE | KL | MSE | KL | MSE | KL | MSE |
| Rand | 12.56 | 0.084 | 14.50 | 0.104 | 15.78 | 0.119 | 16.52 | 0.129 | 16.60 | 0.128 | 16.82 | 0.132 | 16.70 | 0.126 |
| PCA | 2.65 | 0.011 | 5.71 | 0.013 | 7.28 | 0.013 | 7.57 | 0.013 | 6.66 | 0.016 | 6.44 | 0.019 | 6.49 | 0.021 |
| Triplet | 3.19 | 0.014 | 3.61 | 0.014 | 3.97 | 0.015 | 4.14 | 0.015 | 4.16 | 0.015 | 4.18 | 0.015 | 4.33 | 0.015 |
| CPO | 2.22 | 0.010 | 1.80 | 0.008 | 2.32 | 0.012 | 3.27 | 0.014 | 3.81 | 0.015 | 4.13 | 0.015 | 4.44 | 0.015 |
| SimPO | 2.90 | 0.010 | 2.11 | 0.009 | 1.94 | 0.009 | 2.17 | 0.009 | 2.58 | 0.012 | 3.05 | 0.016 | 3.86 | 0.028 |
| ORPO | 2.72 | 0.010 | 1.93 | 0.008 | 1.75 | 0.008 | 1.96 | 0.009 | 2.31 | 0.011 | 2.68 | 0.015 | 3.26 | 0.022 |
| DimPO | **0.67** | **0.005** | **0.97** | **0.006** | **1.29** | **0.007** | **1.57** | **0.008** | **1.83** | **0.009** | **2.10** | **0.011** | **2.62** | **0.014** |

Table 9: Comparison of different projection approaches for Llama-3B-Instruct. We report the KL Divergence of attention weights and the MSE of attention outputs, averaged over all attention layers, for different target dimensions $d' \in \{64, 32, 16, 8, 4, 2, 1\}$.

|  | 64 | | 32 | | 16 | | 8 | | 4 | | 2 | | 1 | |
|---|---|---|---|---|---|---|---|---|---|---|---|---|---|---|
|  | KL | MSE | KL | MSE | KL | MSE | KL | MSE | KL | MSE | KL | MSE | KL | MSE |
| Rand | 14.07 | 1.269 | 14.85 | 1.577 | 15.49 | 1.837 | 15.75 | 2.045 | 15.89 | 2.131 | 16.01 | 2.173 | 15.88 | 1.929 |
| PCA | 4.42 | 0.217 | 6.51 | 0.267 | 8.01 | 0.360 | 9.16 | 0.503 | 10.02 | 0.554 | 10.34 | 0.533 | 10.36 | 0.534 |
| Triplet | 2.50 | 0.149 | 2.90 | 0.155 | 3.14 | 0.159 | 3.39 | 0.165 | 3.54 | 0.166 | 3.64 | 0.167 | 3.75 | 0.169 |
| CPO | 2.52 | 0.216 | 2.06 | 0.170 | 2.54 | 0.148 | 3.12 | 0.156 | 3.44 | 0.165 | 3.55 | 0.169 | 3.79 | 0.168 |
| SimPO | 1.44 | 0.115 | 1.72 | 0.120 | 2.20 | 0.132 | 2.86 | 0.145 | 3.58 | 0.186 | 4.03 | 0.248 | 4.42 | 0.270 |
| ORPO | 1.39 | 0.116 | 1.66 | 0.107 | 2.06 | 0.131 | 2.51 | 0.146 | 2.96 | 0.171 | 3.29 | 0.196 | 3.51 | 0.263 |
| DimPO | **0.73** | **0.052** | **1.15** | **0.076** | **1.57** | **0.089** | **1.89** | **0.107** | **2.07** | **0.117** | **2.27** | **0.132** | **2.74** | **0.223** |

Table 10: Comparison of different projection approaches for Qwen3-4B-Instruct. We report the KL Divergence of attention weights and the MSE of attention outputs, averaged over all attention layers, for different target dimensions $d' \in \{64, 32, 16, 8, 4, 2, 1\}$.

|  | 64 | | 32 | | 16 | | 8 | | 4 | | 2 | | 1 | |
|---|---|---|---|---|---|---|---|---|---|---|---|---|---|---|
|  | KL | MSE | KL | MSE | KL | MSE | KL | MSE | KL | MSE | KL | MSE | KL | MSE |
| Rand | 14.65 | 1.181 | 15.58 | 1.459 | 15.82 | 1.594 | 16.02 | 1.789 | 16.08 | 1.911 | 16.09 | 1.739 | 15.96 | 1.705 |
| PCA | 3.65 | 0.826 | 7.49 | 0.891 | 10.39 | 0.936 | 11.32 | 0.939 | 11.40 | 0.985 | 11.43 | 1.138 | 11.41 | 1.205 |
| Triplet | 2.51 | 0.104 | 2.92 | 0.108 | 3.23 | 0.115 | 3.44 | 0.114 | 3.52 | 0.114 | 3.64 | 0.116 | 3.69 | 0.119 |
| CPO | 2.53 | 0.120 | 2.34 | 0.109 | 2.61 | 0.110 | 3.19 | 0.117 | 3.51 | 0.119 | 3.70 | 0.179 | 3.85 | 0.118 |
| SimPO | 1.58 | 0.094 | 1.83 | 0.101 | 2.43 | 0.120 | 3.18 | 0.144 | 3.96 | 0.174 | 4.44 | 0.193 | 4.63 | 0.254 |
| ORPO | 1.61 | 0.096 | 1.85 | 0.104 | 2.30 | 0.114 | 2.87 | 0.134 | 3.39 | 0.148 | 3.69 | 0.156 | 3.75 | 0.169 |
| DimPO | **0.78** | **0.057** | **1.15** | **0.078** | **1.62** | **0.102** | **2.07** | **0.111** | **2.43** | **0.111** | **2.75** | **0.126** | **3.10** | **0.736** |

Table 11: Comparison of different projection approaches for Qwen2.5-7B-Instruct. We report the KL Divergence of attention weights and the MSE of attention outputs, averaged over all attention layers, for different target dimensions $d' \in \{64, 32, 16, 8, 4, 2, 1\}$.

## C  EFFECT OF KEY-PAIR SELECTION ON PAIRWISE LOSSES

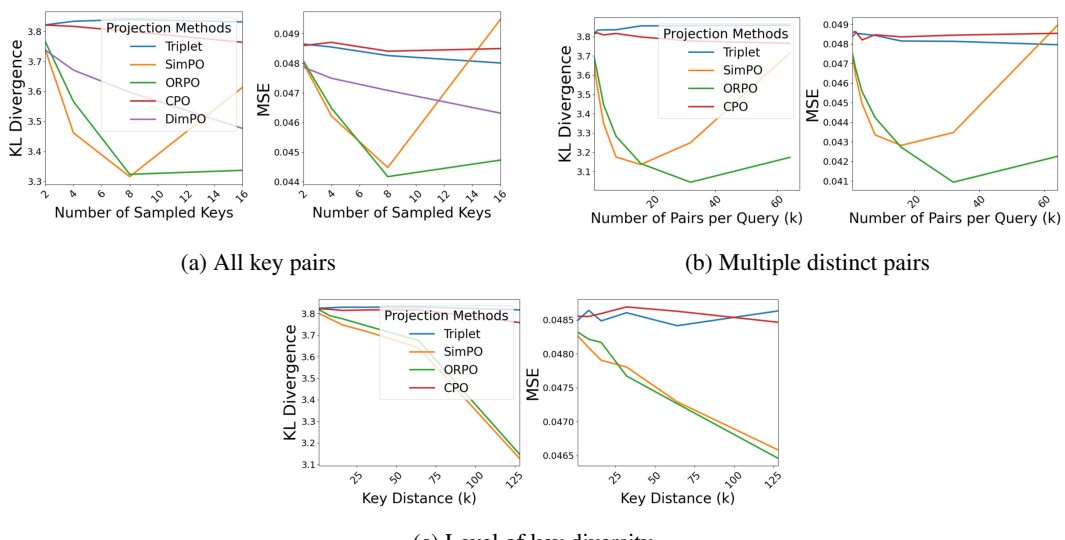

(a) All key pairs

(b) Multiple distinct pairs

(c) Level of key diversity

Figure 4: Effect of key-pair selection on pairwise losses. (a) Using all possible key pairs from a sampled subset shows that DimPO keeps improving as more keys are included, whereas pairwise methods either plateau or degrade beyond a certain point. (b) Training with multiple non-overlapping pairs per query does not yield improvements, suggesting that additional pairs introduce noise rather than meaningful signal. (c) Increasing the diversity between chosen and rejected keys consistently improves performance, indicating that pairwise losses benefit most from highly diverse key pairs.

The observation that DimPO outperforms the other loss functions raises the question of whether this advantage comes from a better inductive bias or simply from receiving more training signal. Although all methods use the same number of training instances, DimPO uses all keys associated with a given query, whereas pairwise losses rely on only two keys (*chosen* and *rejected*). Providing every possible key pair to pairwise methods would be computationally prohibitive due to the combinatorial growth in training examples, but it is still informative to study whether their weaker performance is caused by this information bottleneck. To this end, we perform three controlled experiments on the attention layers of Llama3.2-1B-Instruct, Llama3.2-3B-Instruct, Llama3.1-8B-Instruct, and Qwen3-4B-Instruct, reporting averages across all models. For efficiency, training is performed on 128-token subsequences sampled from 10 chapters, and evaluation uses a validation set of 10 full-length chapters (4096 tokens each) from BOOKSUM.

**All Key Pairs.**  We first investigate pairwise methods and DimPO by sampling $k \in \{2, 4, 8, 16\}$ keys from the 128 available keys per sequence and using all possible key pairs within this subset for

training. Figure 4a reports two metrics, averaged across all attention layers and target dimensions $d' \in \{1, 2, 4, 8, 16, 32, 64\}$ (the same as in the other experiments): KL divergence between the original and projected attention weights, and MSE between the original and projected attention outputs (after applying value vectors). The results indicate that DimPO benefits consistently from having access to more keys, while Triplet and CPO remain largely unaffected. ORPO and SimPO initially seem to gain from additional keys, but their performance quickly plateaus or even degrades, suggesting that the increased combinatorial complexity hinders training rather than helping. This emphasizes that even if pairwise methods were trained on all possible key pairs, they would likely still fall short of DimPO's performance, highlighting the advantage of its listwise formulation.

**Multiple Distinct Pairs.** Given that overlapping keys appear to be a limiting factor, we next investigate training with multiple *distinct* pairs per query, ensuring that no key is used more than once. For each query, we generate $k \in \{1, 2, 4, 8, 16, 32, 64\}$ training pairs, where each *chosen* key comes from the top half of the attention-weight ranking and each *rejected* key from the bottom half. Figure 4b shows that even this distinct-pair setting does not improve performance: pairwise methods consistently perform best when using only a single pair per query, confirming that adding more pairs introduces noise rather than additional useful signal.

**Level of Key Diversity.** Table 1 reports pairwise methods trained by maximizing the diversity between chosen and rejected keys. One might wonder whether using more similar key pairs could be beneficial. In Figure 4c, we show results for $k \in \{1, 8, 16, 32, 64, 127\}$, where $k$ indicates the distance in the attention-weight ranking between chosen and rejected keys (i.e., $k = 1$ corresponds to directly neighboring keys). The results show that while CPO and Triplet losses remain largely unaffected by diversity, SimPO and ORPO exhibit substantial differences across both metrics, highlighting that these methods require highly diverse key pairs to achieve optimal performance.

# D  GENERAL TASK RESULTS

| $d'$ / $\ell$ | 0 | 2 | 4 | 6 | 8 | 10 | 12 | 14 | 16 |
|---|---|---|---|---|---|---|---|---|---|
| **ARC-Challenge** | | | | | | | | | |
| 32 | 37.97 | 37.37 | 36.60 | 36.01 | 30.63 | 29.10 | 27.22 | 23.29 | 23.46 |
| 16 | 37.97 | 37.80 | 35.49 | 33.28 | 28.33 | 26.02 | 25.09 | 23.72 | 22.87 |
| 4 | 37.97 | 37.29 | 32.76 | 29.61 | 25.68 | 23.55 | 23.12 | 25.00 | 25.94 |
| 1 | 37.97 | 36.69 | 33.02 | 27.39 | 23.72 | 24.06 | 23.81 | 23.89 | 26.02 |
| **HellaSwag** | | | | | | | | | |
| 32 | 60.71 | 60.21 | 59.33 | 57.94 | 52.61 | 48.49 | 42.93 | 33.61 | 28.87 |
| 16 | 60.71 | 59.96 | 57.60 | 52.77 | 43.38 | 36.42 | 30.30 | 27.60 | 26.74 |
| 4 | 60.71 | 58.69 | 52.13 | 41.05 | 31.86 | 29.40 | 27.43 | 27.11 | 26.29 |
| 1 | 60.71 | 58.39 | 50.83 | 39.00 | 30.17 | 28.24 | 27.61 | 26.68 | 26.71 |
| **MMLU** | | | | | | | | | |
| 32 | 45.93 | 46.00 | 43.85 | 34.92 | 29.72 | 27.55 | 26.70 | 24.46 | 22.96 |
| 16 | 45.93 | 46.17 | 39.70 | 25.70 | 24.35 | 23.74 | 22.77 | 22.75 | 22.93 |
| 4 | 45.93 | 45.59 | 36.30 | 22.93 | 22.96 | 22.96 | 23.00 | 22.90 | 24.13 |
| 1 | 45.93 | 45.48 | 32.65 | 22.94 | 22.95 | 22.95 | 23.02 | 22.92 | 23.05 |
| **TruthfulQA-mc2** | | | | | | | | | |
| 32 | 43.89 | 44.26 | 44.67 | 43.58 | 43.93 | 45.87 | 46.24 | 47.99 | 50.98 |
| 16 | 43.89 | 43.79 | 43.86 | 45.90 | 47.29 | 49.16 | 50.77 | 50.20 | 48.81 |
| 4 | 43.89 | 44.44 | 46.64 | 50.87 | 50.95 | 50.84 | 50.55 | 49.62 | 48.02 |
| 1 | 43.89 | 44.64 | 47.11 | 50.69 | 50.45 | 50.13 | 50.14 | 49.04 | 48.44 |
| **WinoGrande** | | | | | | | | | |
| 32 | 59.83 | 59.59 | 58.64 | 58.33 | 56.43 | 53.43 | 50.12 | 51.62 | 51.70 |
| 16 | 59.83 | 59.27 | 58.48 | 58.17 | 54.14 | 50.67 | 51.93 | 50.36 | 52.41 |
| 4 | 59.83 | 59.04 | 58.17 | 56.04 | 52.01 | 49.88 | 49.88 | 49.64 | 51.54 |
| 1 | 59.83 | 59.75 | 57.54 | 55.33 | 52.80 | 50.36 | 49.88 | 50.51 | 47.91 |
| **Saved Cache Memory (%)** | | | | | | | | | |
| 32 | 0.00 | 3.12 | 6.25 | 9.38 | 12.50 | 15.62 | 18.75 | 21.88 | 25.00 |
| 16 | 0.00 | 4.69 | 9.38 | 14.06 | 18.75 | 23.44 | 28.12 | 32.81 | 37.50 |
| 4 | 0.00 | 5.86 | 11.72 | 17.58 | 23.44 | 29.30 | 35.16 | 41.02 | 46.88 |
| 1 | 0.00 | 6.15 | 12.30 | 18.46 | 24.61 | 30.76 | 36.91 | 43.07 | 49.22 |

Table 12: Performance and KV cache memory reduction (%) for Llama3.2-1B-Instruct across multiple benchmarks and varying projection dimensions/layers.

| $d' / \ell$ | 0 | 2 | 4 | 7 | 10 | 12 | 14 | 16 | 18 | 21 | 24 | 26 | 28 |
|---|---|---|---|---|---|---|---|---|---|---|---|---|---|
| **ARC-Challenge** | | | | | | | | | | | | | |
| 64 | 45.90 | 46.08 | 45.82 | 45.56 | 43.94 | 43.17 | 41.72 | 40.96 | 38.91 | 37.54 | 34.39 | 31.83 | 25.34 |
| 16 | 45.90 | 43.77 | 44.03 | 42.92 | 35.75 | 31.83 | 28.92 | 27.22 | 26.02 | 25.26 | 23.21 | 23.55 | 26.45 |
| 4 | 45.82 | 43.26 | 40.53 | 35.84 | 29.78 | 25.85 | 25.00 | 22.87 | 23.29 | 24.06 | 24.49 | 25.26 | 27.39 |
| 1 | 45.82 | 43.43 | 38.82 | 36.18 | 27.73 | 25.34 | 24.91 | 24.83 | 24.49 | 24.49 | 24.40 | 24.49 | 27.65 |
| **HellaSwag** | | | | | | | | | | | | | |
| 64 | 70.53 | 70.73 | 70.65 | 70.29 | 69.45 | 69.13 | 67.74 | 65.85 | 64.16 | 61.12 | 57.53 | 53.87 | 48.14 |
| 16 | 70.53 | 70.05 | 67.66 | 65.49 | 57.43 | 54.41 | 47.19 | 41.93 | 37.04 | 32.30 | 28.24 | 27.30 | 26.06 |
| 4 | 70.47 | 69.05 | 63.30 | 53.21 | 41.02 | 34.60 | 30.51 | 29.64 | 28.82 | 27.61 | 26.94 | 26.44 | 26.65 |
| 1 | 70.47 | 68.88 | 61.34 | 50.83 | 37.75 | 31.56 | 28.38 | 27.64 | 27.60 | 26.89 | 26.63 | 26.46 | 26.54 |
| **MMLU** | | | | | | | | | | | | | |
| 64 | 60.38 | 60.02 | 59.98 | 59.14 | 58.67 | 58.62 | 53.38 | 51.00 | 48.50 | 40.86 | 36.54 | 33.06 | 30.34 |
| 16 | 60.38 | 59.76 | 58.29 | 58.46 | 57.15 | 57.30 | 30.47 | 25.94 | 24.69 | 23.56 | 23.27 | 23.56 | 24.28 |
| 4 | 60.50 | 59.89 | 56.72 | 51.69 | 30.81 | 25.25 | 22.92 | 22.92 | 22.92 | 22.92 | 22.96 | 22.96 | 24.87 |
| 1 | 60.50 | 59.71 | 55.10 | 48.95 | 27.97 | 23.47 | 22.95 | 22.94 | 22.94 | 22.94 | 22.94 | 22.94 | 23.48 |
| **TruthfulQA-mc2** | | | | | | | | | | | | | |
| 64 | 49.75 | 50.42 | 50.46 | 49.73 | 49.75 | 49.99 | 49.62 | 48.78 | 48.95 | 48.55 | 46.99 | 47.53 | 48.47 |
| 16 | 49.75 | 50.24 | 51.34 | 50.77 | 49.35 | 48.89 | 51.06 | 50.15 | 49.54 | 49.38 | 49.68 | 50.19 | 48.80 |
| 4 | 49.77 | 50.00 | 51.47 | 52.40 | 49.90 | 47.93 | 47.96 | 48.17 | 48.07 | 48.40 | 48.58 | 48.87 | 48.78 |
| 1 | 49.77 | 50.06 | 51.44 | 52.52 | 49.83 | 48.54 | 47.95 | 47.72 | 47.58 | 47.94 | 48.13 | 48.71 | 49.07 |
| **WinoGrande** | | | | | | | | | | | | | |
| 64 | 67.40 | 68.19 | 67.09 | 67.01 | 67.72 | 68.03 | 66.93 | 65.11 | 62.35 | 58.80 | 55.17 | 55.09 | 52.72 |
| 16 | 67.40 | 67.56 | 65.75 | 65.90 | 64.17 | 63.38 | 61.96 | 57.54 | 52.96 | 51.38 | 48.54 | 49.72 | 51.38 |
| 4 | 67.80 | 67.48 | 65.51 | 63.61 | 61.88 | 59.19 | 57.30 | 53.28 | 50.28 | 50.59 | 48.93 | 49.57 | 49.72 |
| 1 | 67.80 | 66.77 | 65.59 | 62.67 | 60.77 | 56.27 | 55.01 | 51.22 | 50.43 | 50.43 | 47.43 | 47.43 | 48.70 |
| **Saved Cache Memory (%)** | | | | | | | | | | | | | |
| 64 | 0.00 | 1.79 | 3.57 | 6.25 | 8.93 | 10.71 | 12.50 | 14.29 | 16.07 | 18.75 | 21.43 | 23.21 | 25.00 |
| 16 | 0.00 | 3.12 | 6.25 | 10.94 | 15.62 | 18.75 | 21.88 | 25.00 | 28.12 | 32.81 | 37.50 | 40.62 | 43.75 |
| 4 | 0.00 | 3.46 | 6.92 | 12.11 | 17.30 | 20.76 | 24.22 | 27.68 | 31.14 | 36.33 | 41.52 | 44.98 | 48.44 |
| 1 | 0.00 | 3.54 | 7.09 | 12.40 | 17.72 | 21.26 | 24.80 | 28.35 | 31.89 | 37.21 | 42.52 | 46.07 | 49.61 |

Table 13: Performance and KV cache memory reduction (%) for Llama3.2-3B-Instruct across multiple benchmarks and varying projection dimensions/layers.

# E USE OF LARGE LANGUAGE MODELS

For this work, we used GPT-5-mini to help with language polishing, phrasing, and grammar. All scientific content, experimental design, data analysis, and conclusions were developed solely by the authors. We take full responsibility for the final content of this paper, including any text generated with LLM assistance.

| $d' / \ell$ | 0 | 2 | 4 | 8 | 12 | 16 | 20 | 24 | 28 | 30 | 32 |
|---|---|---|---|---|---|---|---|---|---|---|---|
| **ARC-Challenge** | | | | | | | | | | | |
| 64 | 54.95 | 55.29 | 54.78 | 55.38 | 53.16 | 52.05 | 48.81 | 46.50 | 42.49 | 40.87 | 28.84 |
| 16 | 54.95 | 53.67 | 53.16 | 52.39 | 48.12 | 39.76 | 29.95 | 25.85 | 22.53 | 23.38 | 26.02 |
| 4 | 54.95 | 52.30 | 50.60 | 48.12 | 34.56 | 26.37 | 22.53 | 22.35 | 22.44 | 22.95 | 26.19 |
| 1 | 54.95 | 52.13 | 50.51 | 43.52 | 31.31 | 25.17 | 22.78 | 21.59 | 22.95 | 23.81 | 25.60 |
| **HellaSwag** | | | | | | | | | | | |
| 64 | 79.22 | 78.63 | 78.51 | 77.87 | 77.18 | 76.49 | 73.77 | 71.09 | 64.67 | 59.21 | 53.84 |
| 16 | 79.15 | 77.43 | 76.79 | 74.81 | 70.65 | 61.97 | 47.42 | 35.77 | 29.13 | 27.83 | 26.62 |
| 4 | 79.15 | 76.86 | 74.80 | 69.38 | 53.03 | 37.61 | 30.60 | 29.14 | 27.19 | 27.67 | 26.51 |
| 1 | 79.15 | 76.67 | 74.16 | 63.29 | 44.86 | 32.29 | 28.10 | 28.06 | 26.70 | 27.02 | 26.30 |
| **MMLU** | | | | | | | | | | | |
| 64 | 68.12 | 67.75 | 68.08 | 67.69 | 67.62 | 65.84 | 60.23 | 51.62 | 43.93 | 36.85 | 36.38 |
| 16 | 68.02 | 68.03 | 67.75 | 67.74 | 67.63 | 35.71 | 23.08 | 22.97 | 22.97 | 22.91 | 22.94 |
| 4 | 68.02 | 67.59 | 68.02 | 67.23 | 60.90 | 24.59 | 22.96 | 22.89 | 22.95 | 23.10 | 26.27 |
| 1 | 68.02 | 67.75 | 68.09 | 66.16 | 62.24 | 24.26 | 22.95 | 23.03 | 22.95 | 23.24 | 26.26 |
| **TruthfulQA-mc2** | | | | | | | | | | | |
| 64 | 54.00 | 53.93 | 53.90 | 53.86 | 53.90 | 53.91 | 51.09 | 51.37 | 47.24 | 44.28 | 43.49 |
| 16 | 54.07 | 53.80 | 53.87 | 53.61 | 52.36 | 51.65 | 49.43 | 49.87 | 50.28 | 50.27 | 49.02 |
| 4 | 54.07 | 53.63 | 53.77 | 53.68 | 53.13 | 53.08 | 49.93 | 50.31 | 49.71 | 50.61 | 48.85 |
| 1 | 54.07 | 53.56 | 53.76 | 53.16 | 53.08 | 52.91 | 50.22 | 50.36 | 49.47 | 51.22 | 48.98 |
| **WinoGrande** | | | | | | | | | | | |
| 64 | 74.19 | 73.80 | 74.11 | 73.09 | 73.48 | 72.93 | 70.64 | 65.43 | 60.14 | 57.70 | 52.57 |
| 16 | 74.27 | 73.48 | 73.56 | 72.53 | 72.69 | 71.59 | 61.40 | 51.93 | 52.49 | 50.12 | 47.91 |
| 4 | 74.27 | 73.64 | 73.72 | 72.06 | 70.09 | 67.88 | 55.96 | 50.20 | 50.12 | 49.49 | 48.62 |
| 1 | 74.27 | 73.48 | 74.03 | 71.27 | 67.17 | 62.19 | 52.33 | 49.49 | 51.22 | 49.96 | 50.67 |
| **Saved Cache Memory (%)** | | | | | | | | | | | |
| 64 | 0.00 | 1.56 | 3.12 | 6.25 | 9.38 | 12.50 | 15.62 | 18.75 | 21.88 | 23.44 | 25.00 |
| 16 | 0.00 | 2.73 | 5.47 | 10.94 | 16.41 | 21.88 | 27.34 | 32.81 | 38.28 | 41.02 | 43.75 |
| 4 | 0.00 | 3.03 | 6.05 | 12.11 | 18.16 | 24.22 | 30.27 | 36.33 | 42.38 | 45.41 | 48.44 |
| 1 | 0.00 | 3.10 | 6.20 | 12.40 | 18.60 | 24.80 | 31.01 | 37.21 | 43.41 | 46.51 | 49.61 |

Table 14: Performance and KV cache memory reduction (%) for Llama3.1-8B-Instruct across multiple benchmarks and varying projection dimensions/layers.

| $d'$ / $\ell$ | 0 | 2 | 4 | 6 | 9 | 12 | 15 | 18 | 21 | 24 | 27 | 30 | 32 | 34 | 36 |
|---|---|---|---|---|---|---|---|---|---|---|---|---|---|---|---|
| **ARC-Challenge** | | | | | | | | | | | | | | | |
| 64 | 58.19 | 58.19 | 58.70 | 58.62 | 57.25 | 57.51 | 56.14 | 54.78 | 53.92 | 53.16 | 50.43 | 48.46 | 48.38 | 48.89 | 49.15 |
| 16 | 58.19 | 57.08 | 55.97 | 55.12 | 53.41 | 49.32 | 43.26 | 37.12 | 33.79 | 31.23 | 30.55 | 27.39 | 26.45 | 25.34 | 22.61 |
| 4 | 58.19 | 55.72 | 54.61 | 51.11 | 49.06 | 43.00 | 35.41 | 29.78 | 27.39 | 27.30 | 24.57 | 24.57 | 23.55 | 22.53 | 25.43 |
| 1 | 58.45 | 56.23 | 54.35 | 50.51 | 47.27 | 39.68 | 33.87 | 28.92 | 28.16 | 26.79 | 23.98 | 23.81 | 23.63 | 24.49 | 25.00 |
| **HellaSwag** | | | | | | | | | | | | | | | |
| 64 | 69.13 | 69.12 | 68.90 | 68.53 | 68.37 | 68.07 | 67.43 | 66.44 | 65.43 | 63.42 | 60.71 | 59.69 | 58.92 | 58.37 | 57.83 |
| 16 | 69.13 | 66.85 | 65.26 | 63.42 | 61.31 | 58.25 | 53.27 | 48.67 | 44.55 | 41.81 | 37.11 | 32.93 | 32.06 | 31.17 | 27.06 |
| 4 | 69.13 | 66.15 | 64.09 | 60.88 | 57.24 | 50.45 | 42.54 | 38.46 | 35.57 | 33.73 | 30.85 | 29.01 | 28.30 | 27.92 | 25.86 |
| 1 | 69.04 | 66.26 | 63.65 | 60.31 | 56.12 | 48.11 | 40.08 | 36.32 | 33.93 | 32.04 | 29.54 | 28.45 | 27.90 | 27.14 | 26.48 |
| **MMLU** | | | | | | | | | | | | | | | |
| 64 | 70.60 | 70.58 | 70.59 | 70.52 | 70.49 | 69.71 | 66.19 | 63.00 | 59.72 | 55.23 | 48.65 | 49.31 | 48.69 | 48.43 | 48.26 |
| 16 | 70.60 | 70.55 | 70.62 | 70.03 | 70.09 | 51.26 | 32.50 | 26.32 | 24.93 | 24.33 | 23.02 | 23.10 | 23.02 | 23.02 | 22.96 |
| 4 | 70.60 | 70.62 | 70.45 | 69.85 | 68.98 | 31.53 | 24.37 | 23.34 | 23.71 | 23.14 | 23.11 | 23.58 | 23.24 | 23.00 | 23.02 |
| 1 | 70.53 | 70.64 | 70.42 | 69.68 | 68.62 | 28.16 | 24.12 | 23.24 | 23.55 | 23.28 | 23.15 | 24.18 | 23.67 | 23.01 | 23.40 |
| **TruthfulQA-mc2** | | | | | | | | | | | | | | | |
| 64 | 62.63 | 62.65 | 62.78 | 62.86 | 62.64 | 62.26 | 60.70 | 59.50 | 59.49 | 58.57 | 57.02 | 54.62 | 53.47 | 50.98 | 47.78 |
| 16 | 62.63 | 62.31 | 61.72 | 62.41 | 59.88 | 59.81 | 56.64 | 53.52 | 51.43 | 52.12 | 50.58 | 50.59 | 49.95 | 49.70 | 50.10 |
| 4 | 62.63 | 62.16 | 61.41 | 61.40 | 59.33 | 58.50 | 55.07 | 52.98 | 52.63 | 52.30 | 51.25 | 50.76 | 50.68 | 50.48 | 49.63 |
| 1 | 62.64 | 62.14 | 61.44 | 61.40 | 58.87 | 58.02 | 54.51 | 52.53 | 51.94 | 51.87 | 51.74 | 50.80 | 51.30 | 50.86 | 48.05 |
| **WinoGrande** | | | | | | | | | | | | | | | |
| 64 | 67.96 | 68.03 | 67.80 | 68.11 | 67.88 | 67.64 | 66.38 | 65.82 | 64.48 | 62.67 | 62.35 | 60.62 | 61.64 | 60.62 | 58.64 |
| 16 | 67.96 | 66.14 | 65.59 | 66.30 | 65.04 | 64.17 | 61.40 | 57.70 | 55.96 | 53.83 | 53.04 | 52.88 | 49.96 | 51.07 | 46.65 |
| 4 | 67.96 | 66.30 | 65.35 | 65.27 | 64.48 | 61.96 | 57.77 | 52.96 | 52.09 | 50.99 | 49.33 | 49.80 | 47.75 | 47.75 | 50.59 |
| 1 | 68.03 | 65.90 | 65.75 | 64.96 | 63.69 | 61.01 | 56.75 | 51.38 | 51.30 | 49.88 | 49.01 | 50.20 | 49.49 | 48.38 | 51.30 |
| **Saved Cache Memory (%)** | | | | | | | | | | | | | | | |
| 64 | 0.00 | 1.39 | 2.78 | 4.17 | 6.25 | 8.33 | 10.42 | 12.50 | 14.58 | 16.67 | 18.75 | 20.83 | 22.22 | 23.61 | 25.00 |
| 16 | 0.00 | 2.43 | 4.86 | 7.29 | 10.94 | 14.58 | 18.23 | 21.88 | 25.52 | 29.17 | 32.81 | 36.46 | 38.89 | 41.32 | 43.75 |
| 4 | 0.00 | 2.69 | 5.38 | 8.07 | 12.11 | 16.15 | 20.18 | 24.22 | 28.26 | 32.29 | 36.33 | 40.36 | 43.06 | 45.75 | 48.44 |
| 1 | 0.00 | 2.76 | 5.51 | 8.27 | 12.40 | 16.54 | 20.67 | 24.80 | 28.94 | 33.07 | 37.21 | 41.34 | 44.10 | 46.85 | 49.61 |

Table 15: Performance and KV cache memory reduction (%) for Qwen3-4B-Instruct across multiple benchmarks and varying projection dimensions/layers.

| $d' / \ell$ | 0 | 2 | 4 | 7 | 10 | 12 | 14 | 16 | 18 | 21 | 24 | 26 | 28 |
|---|---|---|---|---|---|---|---|---|---|---|---|---|---|
| **ARC-Challenge** | | | | | | | | | | | | | |
| 64 | 55.03 | 54.18 | 54.95 | 54.01 | 51.02 | 49.66 | 48.55 | 49.23 | 48.21 | 48.29 | 47.87 | 46.42 | 46.16 |
| 16 | 55.03 | 50.68 | 50.85 | 47.27 | 41.47 | 39.59 | 38.74 | 34.73 | 34.22 | 31.74 | 30.55 | 27.13 | 24.74 |
| 4 | 55.03 | 54.69 | 53.07 | 47.53 | 40.02 | 37.37 | 36.77 | 33.28 | 30.55 | 25.68 | 25.43 | 23.63 | 25.77 |
| 1 | 55.03 | 55.97 | 53.33 | 47.01 | 39.76 | 37.20 | 35.58 | 31.83 | 31.06 | 25.85 | 26.71 | 23.04 | 25.26 |
| **HellaSwag** | | | | | | | | | | | | | |
| 64 | 80.57 | 80.04 | 79.83 | 79.33 | 77.51 | 76.90 | 76.21 | 75.73 | 75.27 | 74.05 | 73.06 | 72.41 | 71.17 |
| 16 | 80.57 | 79.57 | 78.66 | 75.12 | 69.32 | 65.90 | 59.75 | 54.37 | 51.74 | 44.20 | 41.57 | 34.59 | 26.76 |
| 4 | 80.57 | 79.86 | 78.64 | 73.64 | 65.68 | 61.00 | 54.72 | 49.27 | 46.32 | 36.50 | 33.19 | 29.45 | 26.83 |
| 1 | 80.57 | 79.21 | 77.95 | 72.22 | 63.84 | 58.85 | 52.00 | 47.13 | 44.49 | 33.79 | 31.38 | 28.34 | 26.03 |
| **MMLU** | | | | | | | | | | | | | |
| 64 | 71.76 | 71.81 | 71.49 | 71.44 | 67.33 | 65.52 | 62.55 | 59.48 | 58.03 | 56.12 | 53.63 | 52.86 | 50.55 |
| 16 | 71.76 | 71.69 | 71.47 | 71.15 | 31.69 | 29.33 | 27.28 | 27.05 | 26.70 | 26.44 | 26.33 | 22.75 | 22.96 |
| 4 | 71.76 | 71.71 | 71.34 | 70.92 | 28.04 | 25.70 | 24.41 | 24.40 | 24.37 | 24.18 | 23.21 | 22.94 | 23.27 |
| 1 | 71.76 | 71.77 | 71.27 | 71.02 | 27.47 | 25.21 | 24.29 | 23.94 | 24.15 | 23.49 | 23.13 | 22.92 | 23.17 |
| **TruthfulQA-mc2** | | | | | | | | | | | | | |
| 64 | 64.68 | 64.86 | 64.12 | 64.51 | 58.88 | 58.92 | 58.81 | 59.69 | 59.48 | 56.93 | 56.91 | 56.42 | 54.20 |
| 16 | 64.68 | 64.00 | 62.30 | 63.11 | 57.78 | 55.65 | 51.58 | 51.13 | 51.00 | 50.40 | 52.82 | 52.63 | 48.77 |
| 4 | 64.68 | 64.12 | 61.64 | 62.36 | 57.15 | 53.84 | 48.75 | 48.81 | 46.59 | 48.79 | 52.65 | 50.23 | 48.14 |
| 1 | 64.68 | 64.00 | 61.68 | 62.24 | 56.62 | 53.17 | 48.21 | 48.89 | 47.34 | 47.44 | 52.33 | 50.55 | 47.49 |
| **WinoGrande** | | | | | | | | | | | | | |
| 64 | 71.51 | 69.22 | 70.09 | 70.48 | 65.19 | 64.09 | 64.33 | 60.38 | 58.88 | 60.06 | 59.75 | 56.04 | 57.93 |
| 16 | 71.51 | 67.72 | 67.25 | 66.69 | 61.01 | 56.27 | 56.04 | 53.51 | 52.41 | 49.57 | 51.54 | 48.62 | 48.78 |
| 4 | 71.51 | 68.43 | 67.96 | 67.25 | 59.27 | 56.67 | 52.88 | 52.09 | 52.25 | 50.28 | 51.07 | 49.88 | 51.14 |
| 1 | 71.51 | 68.11 | 67.17 | 66.85 | 59.75 | 57.46 | 53.28 | 51.78 | 52.64 | 49.57 | 50.04 | 49.80 | 49.49 |
| **Saved Cache Memory (%)** | | | | | | | | | | | | | |
| 64 | 0.00 | 1.79 | 3.57 | 6.25 | 8.93 | 10.71 | 12.50 | 14.29 | 16.07 | 18.75 | 21.43 | 23.21 | 25.00 |
| 16 | 0.00 | 3.12 | 6.25 | 10.94 | 15.62 | 18.75 | 21.88 | 25.00 | 28.12 | 32.81 | 37.50 | 40.62 | 43.75 |
| 4 | 0.00 | 3.46 | 6.92 | 12.11 | 17.30 | 20.76 | 24.22 | 27.68 | 31.14 | 36.33 | 41.52 | 44.98 | 48.44 |
| 1 | 0.00 | 3.54 | 7.09 | 12.40 | 17.72 | 21.26 | 24.80 | 28.35 | 31.89 | 37.21 | 42.52 | 46.07 | 49.61 |

Table 16: Performance and KV cache memory reduction (%) for Qwen2-7B-Instruct across multiple benchmarks and varying projection dimensions/layers.

