# OpenReview forum: "DimPO: Dimensionality Reduction for Attention using Preference Optimization"
_ICLR.cc/2026/Conference — Submitted to ICLR 2026_

### Official Review · Reviewer_1Naz · 2025-10-21

**Soundness:** 1
**Presentation:** 2
**Contribution:** 1
**Rating:** 2
**Confidence:** 4

**Summary:**

The paper introduces DimPO, a novel reference-free listwise preference optimization loss designed to reduce the dimensionality of attention key and query vectors in large language models, thereby lowering KV cache memory and computational costs.

**Strengths:**

N/A

**Weaknesses:**

1. Poor performance. The authors need to sacrifice 5% performance when KV cache memory is reduced only by 10%-15%. If we take a look at a contemporary work, native sparse attention [1], they even achieve better performance than standard attention while reducing KV cache memory by at least 50%.
2. Poor performance, especially on longer context. If my understanding is correct, the motivation of saving KV cache is to handle longer context. However, in Table 5, we can see that the longer the sequence is, the more performance the authors have to sacrifice.
3. Limited evaluation. I think the authors could compare their methods with other KV cache saving methods other than preference optimization in their experiments, as they listed in their related work.
4. Efficiency. The authors' method cannot reduce required computation, which is commonly achieved by other KV cache compression methods.

[1]. Native Sparse Attention: Hardware-Aligned and Natively Trainable Sparse Attention. ACL 2025.

**Questions:**

N/A

---

> ### Author Response · Authors · 2025-11-21
>
> Thank you for the feedback. Since no strengths were mentioned, let me summarize two main contributions of the paper:
> 1. Improved dimensionality reduction projection of attention vectors: We present a method to more accurately project attention vectors into lower dimensions, significantly outperforming the current SOTA triplet-loss-based projection from the Mongoose paper. And it is clearly better than random-based projections commonly used for LSH etc...
> 2. New framing of KV cache compression problem: Instead of reducing the number of keys or the number of bits per float, we propose an optimized way to reduce the number of values representing single vectors. Our experiments demonstrate the potential of this approach.
>
> Since there were no questions, we will focus on addressing your concerns:
> - Efficiency and performance compared to other methods: Our method is not competing directly with existing approaches; it optimizes along a different dimension. It is complementary rather than an alternative. In addition to the MagicPIG (which is not directly a kv cache compression but rather sparse attention, yet the effect is the same), we evaluated its performance in combination with other existing KV cache compression methods (SnapKV, KIVI) that we will add into Section 4. Our method not only works well in combination (except KIVI 2 bit) but also improves token throughput compared to applying only existing methods to the base model.
> |Model| RULER avg 4K |tokens/s 4k|
> |:-|-:|--:|
> |base Llama3 8B Instruct|95.07|9.79|
> |DimPO 9.38%|93.64|10.00|
> |KIVI2|94.45|8.85|
> |KIVI2 DimPO 9.38%|90.52|9.17|
> |KIVI4|95.05|8.96|
> |KIVI4 DimPO 9.38%|93.88|10.82|
> |SnapKV1024|95.07|9.97|
> |SnapKV1024 DimPO 9.38%|93.62|10.20|
> |SnapKV2048|95.07|9.92|
> |SnapKV2048 DimPO 9.38%|93.62|10.16|
>
> - Performance with MagicPIG (Table 5): First, the motivation is not only to save KV cache to handle longer context but also to handle larger batch sizes etc.. Second, Table 5 does not aim to show long-context performance alone. It demonstrates how our method can be combined with MagicPIG, illustrating both limitations and potential. The tradeoff depends on the user: DimPO can be run with lower 𝑙, resulting in only a modest performance drop.
> |Model|LONGBENCH|RULER4K|RULER8K|RULER16K|RULER32K|RULER65K|
> |:-|-:|-:|-:|-:|-:|-:|
> |base LLama3.1 8B Instruct|37.83|95.05|93.94|93.39|87.74|84.75|
> |MagicPIG|35.84|92.63|92.35|91.64|86.71|83.67|
> |MagicPIG DimPO 3%|34.96|91.60|90.27|87.90|83.74|81.42|
> |MagicPIG DimPO 6%|33.38|89.95|87.12|83.83|81.83|72.08|
> |MagicPIG DimPO 9%|32.58|87.59|83.41|79.56|75.29|63.66|
>
> - Limited evaluation to preference optimization: Could you clarify this concern? We compared preference optimization in Table 1 with other projection methods (Random, PCA, Triplet-based). Additionally, we included RankNet and ListNet, revealing a new SOTA in dimensionality reduction for attention, outperforming DimPO (see below). Section 3 defines the exact task and corresponding metrics for this dimensionality reduction for attention. Then, Section 4 shows DimPO’s performance on general and long-context tasks. We acknowledge that Section 4 should not be limited to MagicPIG, so we added experiments with other KV cache reduction methods, both individually and in combination with DimPO, showing that, except for KIVI 2-bit, they work well together and DimPO further improves throughput beyond applying other methods to the base model. (as shown in a table above)
>
>
> ## Light reframing of the paper’s narrative:
> We found an even better listwise loss function than the described DimPO - ListNet, showing that listwise losses are crucial for Dimensionality Reduction for Attention (and demonstrating that the triplet-based Mongoose approach is not optimal, which is our goal).
> |KLDivergence|d'=64|
> |:----------------|-----------:|
> |Rand|12.8744173|
> |PCA|3.1014843|
> |Triplet|3.2861550|
> |CPO|2.0677807|
> |SimPO|2.8492284|
> |ORPO|2.6025180|
> |RankNet|2.6680312|
> |DimPO|0.7363492|
> |ListNet|0.2656029|
>
> | MSE|d'= 64 |
> |:--------|----------:|
> | Rand  | 0.0792661 |
> | PCA  | 0.0098033 |
> | Triplet | 0.0109971 |
> | CPO  | 0.0084276 |
> | SimPO | 0.0086739 |
> | ORPO  | 0.0073070 |
> | RankNet | 0.0073032 |
> | DimPO   | 0.0046349 |
> | ListNet | 0.0014229 |
>
> In the end-to-end tasks (for Section 4 in paper), we will show a comparison of different loss functions (listwise as well as pairwise). Here we compare 3 for now, ListNet (listwise) with DimPO (listwise) and ORPO (a well-performing pairwise loss), showing that pairwise methods perform extremely poorly, highlighting the need for listwise losses for dimensionality reduction of attention.
> |Model|RULER avg 4K|tokens/s 4k|
> |:-|--:|--:|
> |SnapKV1024|95.07|9.97|
> |SnapKV1024 DimPO 6.25%|94.27|9.97|
> |SnapKV1024 DimPO 9.38%|93.62|10.20|
> |SnapKV1024 ListNet 6.25%|94.13|9.94|
> |SnapKV1024 ListNet 9.38%|93.65|10.05|
> |SnapKV1024 ORPO 6.25%|65.12|9.95|
> |SnapKV1024 ORPO 9.38%|30.90|10.13|

---

> > ### Comment · Reviewer_1Naz · 2025-11-22
> > **Thanks for the rebuttal.**
> >
> > Although the authors claim that, their method is not competing directly with existing approaches since it optimizes along a different dimension, across all their experiments, we can see clear performance tradeoff even when little (<15%) KV cache is reduced. I do not understand why this direction is necessary, since we already have better solutions. Thus, I keep my score.

---

> > > ### Author Response · Authors · 2025-12-03
> > >
> > > Thank you for the clarification and the feedback. We respectfully offer several points that may address the concern about the necessity of our direction.
> > >
> > > First, IF we evaluate our method purely as a direct KV-cache reduction mechanism, one may indeed argue that a ~10% reduction appears “small”. However, even such reductions can be operationally relevant (e.g., model fitting on a single GPU, increased batch sizes, or avoiding offloading). More importantly, as shown in the comparison table we provided in our previous response, when combined with existing techniques such as SnapKV or KIVI, our projection not only accelerates inference of base model more than SnapKV or KIVI alone, but also increases throughput on top of SnapKV and KIVI, without introducing a significant accuracy drop. This demonstrates another practical utility beyond the ~10% reduction.
> > >
> > > Second, this practical use case is not the primary claim of the paper. Our goal is to show that a learnable listwise projection can preserve model quality surprisingly well and is composable with other compression and KV-reduction techniques. The standalone projection experiment is only one demonstration of its usefulness, not the central contribution.
> > >
> > > Third, compared to prior work on learnable projections-specifically the triplet-loss-based projection proposed by Mongoose for LSH-we observe a substantial improvement. Our approach reduces the KL divergence from **3.2861550 → 0.2656029**, which is a large and non-trivial gain. This directly supports our claim that listwise-optimized projections capture the attention structure much more effectively than existing learnable projections.
> > >
> > > Fourth, to further ground these findings, we provide an extended comparison with several alternative objective functions of direct attention projection during inference on the practical RULER task (will be included in Section 4). The results (shown below) show clearly that listwise methods (DimPO and ListNet) significantly outperform all pairwise losses, and achieve performance where the direct projection alone preserves model quality remarkably well. Even the PO/Ranking pairwise losses consistently outperform the triplet loss, confirming that the problem benefits strongly from moving beyond triplet-based learning.
> > >
> > > | Model                    | RULER avg 4K |
> > > |--------------------------|--------------|
> > > | Llama 8B                 | 95.00 |
> > > | Llama 8B CPO 6.25 %      | 57.58 |
> > > | Llama 8B CPO 9.38 %      | 26.75 |
> > > | Llama 8B CPO 12.50 %     | 5.59 |
> > > | Llama 8B DimPO 6.25 %    | 94.40 |
> > > | Llama 8B DimPO 9.38 %    | 93.77 |
> > > | Llama 8B DimPO 12.50 %   | 90.62 |
> > > | Llama 8B ListNet 6.25 %  | 94.14 |
> > > | Llama 8B ListNet 9.38 %  | 93.76 |
> > > | Llama 8B ListNet 12.50 % | 90.35 |
> > > | Llama 8B ORPO 6.25 %     | 62.16 |
> > > | Llama 8B ORPO 9.38 %     | 30.27 |
> > > | Llama 8B ORPO 12.50 %    | 4.12 |
> > > | Llama 8B RankNet 6.25 %  | 70.36 |
> > > | Llama 8B RankNet 9.38 %  | 38.78 |
> > > | Llama 8B RankNet 12.50 % | 7.80 |
> > > | Llama 8B SimPO 6.25 %    | 63.19 |
> > > | Llama 8B SimPO 9.38 %    | 36.68 |
> > > | Llama 8B SimPO 12.50 %   | 5.66 |
> > > | Llama 8B Triplet 6.25 %  | 12.19 |
> > > | Llama 8B Triplet 9.38 %  | 1.26 |
> > > | Llama 8B Triplet 12.50 % | 0.09 |
> > >
> > > Finally, we do not claim that practitioners “must” use this projection directly as a KV-reduction method. Instead, we argue that - motivated by KV-cache reduction, inference efficiency, and attention compression - this form of projection provides a valuable perspective and a strong learnable building block. It is applicable not only to direct dimensionality-reduced attention, but also naturally integrates with methods based on LSH and other approximate attention mechanisms (and not only). Even if future work adopts this projection primarily as a learnable transformation rather than a direct KV-cache reducer, our findings show that it is both effective and broadly compatible.
> > >
> > > We hope this clarifies why we believe the direction is meaningful and complementary to existing techniques.

---

### Official Review · Reviewer_j6MF · 2025-10-30

**Soundness:** 3
**Presentation:** 3
**Contribution:** 3
**Rating:** 6
**Confidence:** 4

**Summary:**

This paper addresses the memory bottleneck of the Key-Value (KV) cache in Large Language Models (LLMs). The authors propose to reduce the memory footprint by learning a projection to map key and query vectors into a "learned lower-dimensional space" , while leaving value vectors unchanged. The core contribution is twofold: (1) framing this projection task as a "preference optimization problem" , and (2) introducing DimPO, a "novel reference-model-free, listwise preference optimization loss". The paper argues that existing pairwise losses are computationally infeasible and existing listwise losses require a reference model, which is not available in this setting. The DimPO loss function is derived by adapting listwise losses (like LiPO) and removing the reference model by assuming it is a "uniform distribution".

**Strengths:**

1. The paper makes a valuable contribution to an important and active area of research: LLM inference efficiency. The primary contribution is the novel framing of attention dimensionality reduction "as a preference optimization problem". This opens a new avenue for this task.

2. The second contribution, the DimPO loss, is a practical tool derived to fit this new problem formulation. It effectively fills the gap left by existing PO losses. The empirical results, showing a "10-15% reduction in KV cache memory with only about a 5% performance drop on generic tasks", represent a meaningful and practical trade-off for model deployment.

**Weaknesses:**

1. **Training Cost**: The proposed method introduces a significant training overhead for the projection layers. Appendix A reports that DimPO requires "\~5 min/layer" for training, whereas baseline methods like SimPO, Triplet, and ORPO are "substantially faster (\~10 s/layer)". This 30x increase in training time is a significant practical drawback, even if it is a one-time cost.

2. **Weak Long-Context Performance:** The method is strongly motivated by the challenges of "long-context tasks". However, the experimental results on the RULER benchmark (Table 4) are weak for smaller models. The paper notes "Llama 1B... quickly experienced substantial performance degradation". For instance, at 9.38% memory savings, the Llama3.2-1B model's average 4K score drops from 79.35 to 17.72. This severe drop suggests the method, while effective on general tasks, "are more sensitive"  and may not be a viable solution for its primary motivating use case, especially in smaller models.

3. **Issues with Other Methods**: The paper investigates combining DimPO with MagicPIG . The results in Table 5 show a non-trivial drop in performance. The base MagicPIG model achieves 92.63 on RULER 4K, while the "MagicPIG 9.38% (d' = 64, l = 12)" variant scores 87.59. A similar drop is seen on LongBench (35.84 to 32.58). The paper's claim that performance "remains reasonably high"  seems to understate this performance loss. This suggests that the errors from DimPO's projection and MagicPIG's LSH approximation may compound.

4. **Inconsistencies Between the Paper and the Code**:

* 4.1. Definition of the Lambda Weight ($\Delta_{i,j}$)

  * **Paper Description:** The paper, when introducing the LiPO loss framework (Section 3.2), explicitly defines the lambda weight $\Delta_{i,j}$ as:
    $\Delta_{i,j}=|\frac{1}{D(\tau(i))}-\frac{1}{D(\tau(j))}|$
    where $D(\tau(s_{i}))=\log(1+\tau(s_{i}))$.

  * **Code Implementation:** The code implementation in `DimPO/src/models/lowdim_models.py` (inside the `_dimpo_loss` method of the `LowDimDimPO` class) calculates this weight (named `delta`) differently. It includes an additional gain term, $|G_i - G_j|$:

    ```python
    # Gains G_i = 2^{phi_i} - 1
    G = torch.pow(2.0, psi) - 1.0               # [B, K]

    # Discounts D(tau(i)) = log(1 + tau(i))
    D = torch.log1p(ranks.float())              # [B, K]

    # ... (omitted code) ...

    # Pairwise differences
    # ...
    G_i = G.unsqueeze(2)
    G_j = G.unsqueeze(1)
    delta_G = torch.abs(G_i - G_j) # [B, K, K]

    D_i = D.unsqueeze(2)
    D_j = D.unsqueeze(1)
    delta = delta_G * (1.0 / D_i - 1.0 / D_j) # [B, K, K]
    ```

  * **Discrepancy:** The code implements the weight as $\Delta_{i,j} = |G_i - G_j| \cdot |\frac{1}{D(\tau(i))} - \frac{1}{D(\tau(j))}|$. The gain term $|G_i - G_j|$ (where $G_i = 2^{\psi_i} - 1$ is defined in the paper) is present in the code but is omitted from the explicit definition of $\Delta_{i,j}$ provided in the paper.

---

* 4.2. Sign of the Final Loss Function (Equation 1)

  * **Paper Description:** Equation 1 defines the DimPO loss with a leading negative sign:
    $\mathcal{L}_{DimPO}(\pi_{\theta})=-\mathbb{E}_{(x,y,\psi)\sim\mathcal{D}}[\sum_{\psi_{i}>\psi_{j}}\Delta_{i,j}\log(1+e^{-(s_{i}-s_{j}-\gamma)})]$

  * **Code Implementation:** The code implementation in `DimPO/src/models/lowdim_models.py` (in the `_dimpo_loss` method) computes a positive loss value, which is then minimized by the optimizer:

    ```python
    # pair_loss = log(1 + exp(-(s_i - s_j - gamma)))
    pair_loss = F.softplus(-s_diff)             # [B, K, K]

    # Apply mask and weight
    weighted = (delta * pair_loss) * mask       # [B, K, K]
    loss_per_list = weighted.sum(dim=(1,2)) / mask.sum(dim=(1,2)).clamp_min(1)     #[B]

    return loss_per_list.mean()
    ```

  * **Discrepancy:** The code calculates a positive loss (the mean of the listwise softplus terms, weighted by $\Delta_{i,j}$) and returns this positive value for minimization. However, the paper's Equation 1 includes a leading negative sign. Minimizing the paper's formula as written would be equivalent to *maximizing* the positive loss term calculated in the code.

    This strongly suggests the negative sign in Equation 1 is a typo, as the code's implementation (minimizing a positive loss) represents the standard and mathematically correct optimization objective.

**Questions:**

1. The performance collapse of Llama3.2-1B on RULER (Table 4) is severe (e.g., 79.35 -> 17.72). The paper states larger models are "more robust". Can the authors provide insight into why this degradation is so extreme for smaller models? Does this imply that the learned projection fails to capture essential long-range attention patterns in models with lower capacity, and does this limit the method's applicability to only very large (>7B) models for long-context tasks?

2. Regarding the 30x training time increase for DimPO ("\~5 min/layer") compared to baselines ("\~10 s/layer") (Appendix A) : Is this cost primarily from the listwise loss computation, which must consider $O(m^2)$ pairs (where $m=4096$ in training)? Given this high cost, do the authors believe the improved KL/MSE (Table 1) over the much-faster ORPO/SimPO baselines justifies this trade-off?

3. In the MagicPIG combination experiment (Table 5), the authors applied DimPO "to the last 12 layers", resulting in a significant performance drop (e.g., 92.63 -> 87.59 on RULER 4K). Why were 12 layers chosen for this experiment? Could the observed performance drop be due to compounding approximation errors (LSH from MagicPIG and projection from DimPO)? Would applying DimPO to fewer layers (e.g., $l=4$ or $l=8$) have achieved a better balance of memory savings and performance preservation?

4. The derivation in Section 3.2 removes the reference model $\pi_\text{ref}$ by "treat[ing] $\pi_{ref}$ as a uniform distribution". The paper also cites SimPO, which "argues that using a reference model... is inconsistent with inference" , and adopts its margin $\gamma$. Can the authors please clarify the precise novelty of DimPO's formulation compared to what might be termed a "Listwise SimPO" (i.e., applying SimPO's reference-free reward $s_i = \beta \log \pi_\theta(y_i|x)$ directly to the LiPO listwise loss)?

---

> ### Author Response · Authors · 2025-11-21
>
> Thank you for the feedback and additional ideas.
>
> ## Answers to your questions:
> 1. Based on Tables 1 and 8-11, model size does not significantly affect projection quality. LLaMA 1B performs slightly worse, likely because its original vectors are 64-dimensional instead of 128, making 64->32 compression harder than 128->64, even though both reduce the same KV cache percentage.But as Table 4 shows, performance does not drop to 0 at 6.25% or 9.38% reduction showing it is more likely just more sensitive because Llama 1B is already well-compressed and more sensitive to further reduction. Base Llama 1B Instruct also performs poorly on the RULER task in general. So basically our method targets larger models anyway (≥4B), which are 1. more likely to need KV cache reduction 2. well performing on those tasks in their base settings.
> 2. Overhead is not from the loss computation itself, but from computing scores (probabilities of the linear layer of given key being similar to the given query), which require attention weights of all projected key vectors and the query token (lines 197–199). This tradeoff is justified, as pairwise losses are unusable for long-context tasks (we will report it for more loss functions in Section 4, for now, see the comparison of ORPO vs DimPO in combination with SnapKV):
> |Model|RULER avg 4K|tokens/s 4k|
> |:--|-:|--:|
> |SnapKV1024|95.07|9.97|
> |SnapKV1024 DimPO 6.25%|94.27|9.97|
> |SnapKV1024 DimPO 9.38%|93.62|10.20|
> |SnapKV1024 ORPO 6.25%|65.12|9.95|
> |SnapKV1024 ORPO 9.38%|30.90|10.13|
>
> 3. MagicPIG–DimPO tradeoff:
> |Model|LONGBENCH|RULER4K|RULER8K|RULER16K|RULER32K|RULER65K|
> |:-|-:|-:|-:|-:|-:|-:|
> |base LLama3.1 8B Instruct|37.83|95.05|93.94|93.39|87.74|84.75|
> |MagicPIG|35.84|92.63|92.35|91.64|86.71|83.67|
> |MagicPIG DimPO 3%|34.96|91.60|90.27|87.90|83.74|81.42|
> |MagicPIG DimPO 6%|33.38|89.95|87.12|83.83|81.83|72.08|
> |MagicPIG DimPO 9%|32.58|87.59|83.41|79.56|75.29|63.66|
>
> 3. DimPO novelty is exactly the reference model-free listwise PO loss motivated by dimensionality reduction for attention (or as you described it listwise SimPO). It also includes the equation often called the model’s policy (lines 197–199) via attention weights of projected vectors. However, we found an even better listwise loss (see below), so Section 3.2 will move to the appendix and the spotlight will move to listwise losses in general (see below).
>
> ## To address your concerns:
> 1. Post-training cost: As you noted, this is a one-time step. Additionally, training can be parallelized since the training of each layer is independent of the others.
> 2. Long-context performance on smaller models: As mentioned above, our focus is on optimizing larger and generally more robust models, rather than already compressed small models.
> 3. DimPO in combination with other methods: In addition to combining with MagicPIG, we are extending the comparison to include SnapKV and KIVI (to be included in Section 4):
> |Model| RULER avg 4K |tokens/s 4k|
> |:-|-:|--:|
> |base Llama3 8B Instruct|95.07|9.79|
> |DimPO 9.38%|93.64|10.00|
> |KIVI2|94.45|8.85|
> |KIVI2 DimPO 9.38%|90.52|9.17|
> |KIVI4|95.05|8.96|
> |KIVI4 DimPO 9.38%|93.88|10.82|
> |SnapKV1024|95.07|9.97|
> |SnapKV1024 DimPO 9.38%|93.62|10.20|
> |SnapKV2048|95.07|9.92|
> |SnapKV2048 DimPO 9.38%|93.62|10.16|
> 4. Code-paper inconsistencies: Thank you for catching these! Both are typos in the paper; the code is correct. We will fix the paper accordingly.
>
>
> ## Light reframing of the paper’s narrative:
> We found an even better listwise loss function than the described DimPO - ListNet, showing that listwise losses are crucial for Dimensionality Reduction for Attention (and demonstrating that the triplet-based Mongoose approach is not optimal, which is our goal).
> |KLDivergence|d'=64|
> |:----------------|-----------:|
> |Rand|12.8744173|
> |PCA|3.1014843|
> |Triplet|3.2861550|
> |CPO|2.0677807|
> |SimPO|2.8492284|
> |ORPO|2.6025180|
> |RankNet|2.6680312|
> |DimPO|0.7363492|
> |ListNet|0.2656029|
>
> | MSE|d'= 64 |
> |:--------|----------:|
> | Rand  | 0.0792661 |
> | PCA  | 0.0098033 |
> | Triplet | 0.0109971 |
> | CPO  | 0.0084276 |
> | SimPO | 0.0086739 |
> | ORPO  | 0.0073070 |
> | RankNet | 0.0073032 |
> | DimPO   | 0.0046349 |
> | ListNet | 0.0014229 |
>
> In the end-to-end tasks (for Section 4 in paper), we will show a comparison of different loss functions (listwise as well as pairwise). Here we compare 3 for now, ListNet (listwise) with DimPO (listwise) and ORPO (a well-performing pairwise loss), showing that pairwise methods perform extremely poorly, highlighting the need for listwise losses for dimensionality reduction of attention.
> |Model|RULER avg 4K|tokens/s 4k|
> |:-|--:|--:|
> |SnapKV1024|95.07|9.97|
> |SnapKV1024 DimPO 6.25%|94.27|9.97|
> |SnapKV1024 DimPO 9.38%|93.62|10.20|
> |SnapKV1024 ListNet 6.25%|94.13|9.94|
> |SnapKV1024 ListNet 9.38%|93.65|10.05|
> |SnapKV1024 ORPO 6.25%|65.12|9.95|
> |SnapKV1024 ORPO 9.38%|30.90|10.13|

---

> > ### Comment · Reviewer_j6MF · 2025-11-21
> > **Thank you!**
> >
> > Thank you for the author's clarification and detailed explanation; all my questions have been adequately answered. I will maintain this weak accept score of 6, and I will continue to follow your discussions with the other reviewers (I'd like to see your answers to some other questions I'm also curious about), and I reserve the possibility of raising the score to 8.
> >
> > Good luck!

---

### Official Review · Reviewer_4Q9G · 2025-10-31

**Soundness:** 3
**Presentation:** 3
**Contribution:** 3
**Rating:** 6
**Confidence:** 4

**Summary:**

The paper proposes DimPO, a reference-free list-wise preference-optimization loss that learns a single linear projection to compress query/key vectors while keeping the attention distribution essentially unchanged. Applied to the last few layers of LLaMA-3 (1\~8 B) and Qwen-2.5/3 (4\~7 B) models, DimPO yields 10–15 % KV-cache memory reduction with ≤5 % average drop on five short-context benchmarks and comparable or slightly larger degradation on long-context suites. It outperforms triplet, PCA, random-projection, and existing pairwise preference losses (SimPO, ORPO, CPO) by 2–3× in KL divergence between original and projected attention weights.

**Strengths:**

1. Novel framing: first to treat attention dimensionality reduction as a preference-ranking problem rather than hashing or reconstruction, motivating a natural list-wise objective.

2. DimPO loss is elegantly reference-free, avoids the quadratic explosion of pairwise comparisons, and consistently beats strong baselines.

3. Thorough experimental sweep: ablates dimension, number of projected layers, and model scale; reports both attention-quality metrics (KL, MSE) and downstream task scores; includes long-context benchmarks and throughput measurements.

4. Practical impact: a 10% cache cut translates directly into larger batch sizes or longer contexts for serving; the projection adds only a small linear layer with negligible latency.

**Weaknesses:**

1. Scope of evaluation: all experiments compress only the last few layers; no systematic study of how layer depth interacts with sensitivity (only a top-down heuristic).

2. Task coverage: long-context evaluation is restricted to 4k and 8k RULER tasks and up to 65 k LongBench; no examination of >100 k tokens or code/documentation datasets where attention patterns differ.

3. Baseline omissions: no comparison against simultaneous KV-quantization (e.g., 4-bit KIVI) or low-rank factorization of the full attention matrix leaving the combined settings unclear.

**Questions:**

1. Why restrict the projection to a single shared linear matrix per layer? Did you try head-specific or unshared version that might yield higher compression with similar parameter count?

2. How does DimPO behave when combined with 4-bit KV quantization—do the errors compound multiplicatively?

3. Did you explore non-uniform layer-wise allocation of dimensions (e.g., d'=32 for early layers, d'=8 for final ones) to push compression further where attention is most robust?

4. Are there attention-pattern failure modes (e.g., many-shot retrieval, long code files) where DimPO degrades catastrophically?

---

> ### Author Response · Authors · 2025-11-21
>
> Thank you for the feedback and additional ideas.
>
> ## Answers to your questions:
> 1. There are essentially two reasons. The first and main one is that we primarily build on the Mongoose paper, where they also use exactly one triplet-based projection shared across all heads. The second reason is that training a separate projection for every layer would be very expensive in terms of post-training time. In our preliminary experiments on a 1-layer transformer, we observed that whether the projection is shared across all heads or not does not make a difference (if anything, sharing seemed to help reduce overfitting). However, it is true that this effect should be measured more precisely on end-to-end tasks for a clearer picture.
> 2. KIVI 4-bit with DimPO works very well, and the errors do not compound - we only observe the inherent lossiness of DimPO when ℓ becomes larger. In contrast, KIVI 2-bit does not work well together with DimPO.
> |Model|RULER avg4K|RULER avg8K|tokens/s4k|tokens/s8k|
> |:---|----:|-----:|---:|----:|
> |base Llama3.1 8B Instruct|95.07|93.95|9.79|3.00|
> |KIVI2|94.45|92.64|8.85|3.90|
> |KIVI2 DimPO 6.25%|91.94|86.40|8.99|4.24|
> |KIVI2 DimPO 9.38%|90.52|83.07|9.17|4.44|
> |KIVI2 DimPO 12.50%|81.17|71.05|9.87|5.14|
> |KIVI4|95.05|93.95|8.96|3.93|
> |KIVI4 DimPO 6.25%|94.36|91.35|8.81|3.99|
> |KIVI4 DimPO 9.38%|93.88|89.65|10.82|4.15|
> |KIVI4 DimPO 12.50%|90.33|85.08|11.18|4.34|
> 3. Yes, in theory the alternative setup could make sense, but in practice we obtain the same performance for a smaller KV-cache reduction. For example, on HellaSwag: layers 25-31 with d’=64 and layer 32 with d’=1 (7% kv cache reduction) gives score 76. This is the same performance as using d’=64 on layers 17–32 (13% kv cache reduction), but the latter gives a twice better reduction ratio. Similarly, 29:d’=8, 30:d’=16, 31:d’=32, 32:d’=64 (5%) performs the same as layers 17-32 with d’=64 (13%).
> 4. Nothing special happens - it works very similarly to the full base model. Do you have any specific benchmarks on your mind you are interested in?
>
> ## To address your concerns:
> - Why a top-down heuristic? A very good point. Our argument is that, as explained in lines 368–370, this is intuitively the most naive way to reduce potential error propagation. We measured each individual layer, and the closer the layer is to the beginning, the more sensitive it is to modification (likely due to error accumulation). Therefore, the safest approach is to reduce the KV cache from the top layers. We are not sure there is space in the main paper for a full evaluation of ℓ=1 or ℓ=4 across different layers - this would likely fit better in the Appendix.
> - 100k+-token tasks: A comparison on >100k tokens would indeed be interesting, but we unfortunately do not have sufficient computational resources. As mentioned in lines 430–431, we must use the eager attention implementation because the shapes (dimensions) of the query, key, and value vectors differ, which is not supported by flash attention or optimized by SDPA.
> - Combined settings comparison: We will extend the results in Section 4 (in addition to MagicPIG's combination) with the following combined settings of DimPO with SnapKV (1024, 2048) and KIVI (2-bit, 4-bit).
> |Model| RULER avg 4K |tokens/s 4k|
> |:-|-:|--:|
> |base Llama3 8B Instruct|95.07|9.79|
> |DimPO 9.38%|93.64|10.00|
> |KIVI2|94.45|8.85|
> |KIVI2 DimPO 9.38%|90.52|9.17|
> |KIVI4|95.05|8.96|
> |KIVI4 DimPO 9.38%|93.88|10.82|
> |SnapKV1024|95.07|9.97|
> |SnapKV1024 DimPO 9.38%|93.62|10.20|
> |SnapKV2048|95.07|9.92|
> |SnapKV2048 DimPO 9.38%|93.62|10.16|
>
> ## Light reframing of the paper’s narrative:
> We found an even better listwise loss function than the described DimPO - ListNet, showing that listwise losses are crucial for Dimensionality Reduction for Attention (and demonstrating that the triplet-based Mongoose approach is not optimal, which is our goal).
> |KLDivergence|d'=64|
> |:----------------|-----------:|
> |Rand|12.8744173|
> |PCA|3.1014843|
> |Triplet|3.2861550|
> |CPO|2.0677807|
> |SimPO|2.8492284|
> |ORPO|2.6025180|
> |RankNet|2.6680312|
> |DimPO|0.7363492|
> |ListNet|0.2656029|
>
> | MSE|d'= 64 |
> |:--------|----------:|
> | Rand  | 0.0792661 |
> | PCA  | 0.0098033 |
> | Triplet | 0.0109971 |
> | CPO  | 0.0084276 |
> | SimPO | 0.0086739 |
> | ORPO  | 0.0073070 |
> | RankNet | 0.0073032 |
> | DimPO   | 0.0046349 |
> | ListNet | 0.0014229 |
>
> In the end-to-end tasks (for Section 4 in paper), we will show a comparison of different loss functions (listwise as well as pairwise). Here we compare 3 for now, ListNet (listwise) with DimPO (listwise) and ORPO (a well-performing pairwise loss), showing that pairwise methods perform extremely poorly, highlighting the need for listwise losses for dimensionality reduction of attention.
> |Model|RULER avg 4K|tokens/s 4k|
> |:-|--:|--:|
> |SnapKV1024|95.07|9.97|
> |SnapKV1024 DimPO 6.25%|94.27|9.97|
> |SnapKV1024 DimPO 9.38%|93.62|10.20|
> |SnapKV1024 ListNet 6.25%|94.13|9.94|
> |SnapKV1024 ListNet 9.38%|93.65|10.05|
> |SnapKV1024 ORPO 6.25%|65.12|9.95|
> |SnapKV1024 ORPO 9.38%|30.90|10.13|

---

### Official Review · Reviewer_ij1L · 2025-10-31

**Soundness:** 2
**Presentation:** 2
**Contribution:** 2
**Rating:** 2
**Confidence:** 4

**Summary:**

This paper introduces a learned dimensionality reduction technique for keys in the KV cache of attention layers, aiming to reduce KV cache memory usage.

**Strengths:**

The proposed approach achieves a 10–15% reduction in KV cache memory.

**Weaknesses:**

1.	Limited evaluation: Reducing KV cache memory is a well-studied topic. Prior work has explored both reducing the number of keys and reducing key dimensions. The paper primarily compares against projection-based key dimension reduction methods, which makes the evaluation less comprehensive and less convincing.
2.	Missing key objectives: Most KV cache optimization techniques ultimately aim to improve inference speed or throughput. The paper does not report comparison results on these critical metrics, leaving the practical impact unclear.
3.	Evaluation metrics: Using KL divergence and MSE to compare projection methods (Table 1) is questionable because the ultimate goal is to maintain end-to-end model performance. Direct comparisons on end-to-end performance would be more informative.
4.	Incomplete comparison of pairwise losses: There are several established pairwise ranking losses (e.g., RankNet, ListNet) designed to preserve ranking. Including these in the comparison would provide a clearer picture of the proposed method’s advantages.
5.	Performance concern: Table 5 shows that applying DimPO on top of existing KV cache optimization methods introduces a noticeable performance drop, which raises concerns about its practical utility.

**Questions:**

1.	In line 181, should F(K_{k,q}) actually be F(K_{l,q})?
2.	What is the additional inference cost introduced by DimPO?

---

> ### Author Response · Authors · 2025-11-21
>
> Thank you for the feedback and additional ideas.
>
> ## Answers to your questions:
> 1. Yes - the correct expression is F(K_{l,q}).
> 2. The only inference-time cost is the linear projection layer; this is negligible relative to the acceleration in attention. Training this layer is a one-time post-training step taking 3–5 minutes per layer (Appendix A, L762).
>
> ## To address your concerns with respect to what is already mentioned in the first version of the paper:
> 1. Missing key objectives: the throughput metric you mention is reported in Table 4, showing the speed-up achieved by our method.
> 2. Evaluation metrics: you point out that KL-divergence and MSE may be insufficient. This seems to be a presentation issue on our side - the paper is effectively divided into two parts: Section 3 focuses on dimensionality reduction for attention, directly defining the task and analyzing its behavior - including the formal definition of the task we are dealing with where KL-divergence and MSE seems to be ideal metrics for that (this part is important to understand how the method works in "background"), while Section 4 then compares, as you note, the end-to-end results, currently only for DimPO as our best method so far.
> 3. Performance concern: The method does not have to be combined with specifically with MagicPIG so the perfrmance drop is not that large and still it can reduce kv cache size and enable bigger batch sizes or bigger context. However still, from Table 5 it may appear that it is not that practical for combining MagicPIG with DimPO for long long long contexts - of course, the tradeoff depends on how many layers one is willing to replace; replacing fewer layers yields better performance with a smaller KV-cache reduction. See the following table:
> |Model|LONGBENCH|RULER4K|RULER8K|RULER16K|RULER32K|RULER65K|
> |:-|-:|-:|-:|-:|-:|-:|
> |base LLama3.1 8B Instruct|37.83|95.05|93.94|93.39|87.74|84.75|
> |MagicPIG|35.84|92.63|92.35|91.64|86.71|83.67|
> |MagicPIG DimPO 3%|34.96|91.60|90.27|87.90|83.74|81.42|
> |MagicPIG DimPO 6%|33.38|89.95|87.12|83.83|81.83|72.08|
> |MagicPIG DimPO 9%|32.58|87.59|83.41|79.56|75.29|63.66|
>
>
> ## Based on your feedback, we would like to present new results and the direction we are going to take:
> - We would like to slightly reframe the paper as “Dimensionality Reduction for Attention using Listwise Losses”, or something along these lines.
> - RankNet and ListNet are simplified forms of PO losses (and are in fact more like frameworks than specific losses). However, once inserted into the comparison table, we can see that listwise losses significantly outperform pairwise losses (which is the goal, since our work builds primarily on Mangoose paper, which uses triplet-based projection - something we want to show is suboptimal), demonstrating that ListNet is a new SOTA:
>
>  - In the context of end-to-end evaluation in practical settings and comparison of our method with the others - here it is important to clarify that we do not propose a competing method to H2O, SnapKV, KIVI, etc., but rather a complementary one (as already compared in Table 5 with MagicPIG). We now add results comparing their performance (currently DimPO; for the final results we will report ListNet, which behaves similarly, slightly better):
> |KLDivergence|d'=64|
> |:-|-:|
> |Rand|12.8744173|
> |PCA|3.1014843|
> |Triplet|3.2861550|
> |CPO|2.0677807|
> |SimPO|2.8492284|
> |ORPO|2.6025180|
> |RankNet|2.6680312|
> |DimPO|0.7363492|
> |ListNet|0.2656029|
>
>  - In the context of end-to-end evaluation in practical settings and comparison of our method with the others - here it is important to clarify that we do not propose a competing method to H2O, SnapKV, KIVI, etc., but rather a complementary one (as already compared in Table 5 with MagicPIG). We now add results comparing their performance (currently DimPO; for the final results we will report ListNet, which behaves similarly, sometimes slightly better) - (not enough of character limit for 8k context results):
> |Model| RULER avg 4K |tokens/s 4k|
> |:-|-:|--:|
> |base Llama3 8B Instruct|95.07|9.79|
> |DimPO 9.38%|93.64|10.00|
> |KIVI2|94.45|8.85|
> |KIVI2 DimPO 9.38%|90.52|9.17|
> |KIVI4|95.05|8.96|
> |KIVI4 DimPO 9.38%|93.88|10.82|
> |SnapKV1024|95.07|9.97|
> |SnapKV1024 DimPO 9.38%|93.62|10.20|
> |SnapKV2048|95.07|9.92|
> |SnapKV2048 DimPO 9.38%|93.62|10.16|
>
> - For comparing different loss functions on end-to-end tasks, DimPO and ListNet behave similarly, while ORPO (paper will include  additional pairwise losses) performs very poorly, clearly demonstrating the need for listwise losses for dimensionality reduction:
>
> |Model|RULER avg 4K|tokens/s 4k|
> |:--|-:|--:|
> |SnapKV1024|95.07|9.97|
> |SnapKV1024 DimPO 6.25%|94.27|9.97|
> |SnapKV1024 DimPO 9.38%|93.62|10.20|
> |SnapKV1024 ListNet 6.25%|94.13|9.94|
> |SnapKV1024 ListNet 9.38%|93.65|10.05|
> |SnapKV1024 ORPO 6.25%|65.12|9.95|
> |SnapKV1024 ORPO 9.38%|30.90|10.13|
>
> - We also observe that the acceleration is larger (using float16 and eager attention) than in the mentioned methods on the same base model.

---

> > ### Author Response · Authors · 2025-12-03
> >
> > To address the concern that "Direct comparisons on end-to-end performance would be more informative." we provide a detailed comparison of different learnable projection approaches - including preference-optimization losses, ranking losses, and triplet loss - on the end-to-end RULER 4K. In addition to KL and MSE metrics, we now report end-to-end task performance (will be included in section 4). The results show that listwise losses (DimPO, ListNet) substantially outperform all pairwise losses (CPO, SimPO, RankNet, ORPO), while the prior-work triplet objective performs extremely poorly. This demonstrates that listwise formulations are the only ones that reliably preserve attention behavior well enough to be usable for direct attention projection during inference. Unlike triplet-based approaches designed primarily for LSH (e.g., Mongoose), listwise methods produce projections that are accurate enough to be applied as real low-dimensional attention replacements.
> >
> > | Model                    |   avg 4K |
> > |:-------------------------|---------:|
> > | Llama 8B                 |    95.00 |
> > | Llama 8B CPO 6.25 %      |    57.58 |
> > | Llama 8B CPO 9.38 %      |    26.75 |
> > | Llama 8B CPO 12.50 %     |     5.59 |
> > | Llama 8B DimPO 6.25 %    |    94.40 |
> > | Llama 8B DimPO 9.38 %    |    93.77 |
> > | Llama 8B DimPO 12.50 %   |    90.62 |
> > | Llama 8B ListNet 6.25 %  |    94.14 |
> > | Llama 8B ListNet 9.38 %  |    93.76 |
> > | Llama 8B ListNet 12.50 % |    90.35 |
> > | Llama 8B ORPO 6.25 %     |    62.16 |
> > | Llama 8B ORPO 9.38 %     |    30.27 |
> > | Llama 8B ORPO 12.50 %    |     4.12 |
> > | Llama 8B RankNet 6.25 %  |    70.36 |
> > | Llama 8B RankNet 9.38 %  |    38.78 |
> > | Llama 8B RankNet 12.50 % |     7.80 |
> > | Llama 8B SimPO 6.25 %    |    63.19 |
> > | Llama 8B SimPO 9.38 %    |    36.68 |
> > | Llama 8B SimPO 12.50 %   |     5.66 |
> > | Llama 8B Triplet 6.25 %  |    12.19 |
> > | Llama 8B Triplet 9.38 %  |     1.26 |
> > | Llama 8B Triplet 12.50 % |     0.09 |

---

### Official Review · Reviewer_Yw4L · 2025-11-01

**Soundness:** 2
**Presentation:** 3
**Contribution:** 2
**Rating:** 4
**Confidence:** 4

**Summary:**

The paper introduces DimPO (Dimensionality Reduction for Attention using Preference Optimization), a post-training method aimed at reducing the KV cache size. The core idea is to learn a linear projection that maps the key (K) and query (Q) vectors into a lower-dimensional space. To achieve this, the authors propose a DimPO frames the task as a preference optimization problem, introducing a reference-model-free, listwise loss function that optimizes the projection matrices to ensure the compressed attention distribution closely matches the original, full-dimensional distribution. Experiments show that DimPO successfully reduces KV cache memory by 10-15% while retaining high performance (over 95%) on general benchmarks and maintaining competitive results on demanding long-context tasks.

**Strengths:**

- The proposed method poses the problem of dimensionality reduction in query and key to the preference optimization (PO) problem. The authors consider the original attention scores as the "preferred" ranking and the compressed attention scores as the "current" ranking. They propose the DimPO loss which directly optimizes the relative order of attention weights. This is fundamentally more effective for preserving the critical attention distribution than simply minimizing the absolute distance between vectors.

- The paper compares DimOP to other preference optimizations including Triplet, CPO, SimPO and ORPO, and shows that DimPO outperforms in terms of KL and MSE on attention weights.

- The method provides memory reduction on KV cache. When the dimensionality reduces from 128 to 64, the paper shows that DimPO can achieve a 10-15% memory reduction.

**Weaknesses:**

- DimPO requires an additional post-training stage to learn the optimal low-dimensional projection. This introduces an additional cost and time compared to post-hoc optimization methods that can be applied directly to a pre-trained model without further training. In addition, learning the projection depends on the quality and diversity of the training dataset used for fine-tuning, potentially requiring high-quality, long-context-specific data to generalize well.

- The primary goal of the proposed method is to reduce the KV cache memory size for efficient token generation in LLM. However, the paper does not provide a direct head-to-head benchmark against other common KV cache optimization techniques like H2O (heavy-hitter eviction), SnapKV (token selection), or KIVI (quantization), which are mentioned in the related works section. Beyond the token eviction method, quantization is a dimensionality reduction approach, and there is a recent work [1] that applies a signed random projection on QK to reduce KV cache. It would be good to compare those works to the proposed method.

[1] Zandieh, Amir, Majid Daliri, and Insu Han. "Qjl: 1-bit quantized jl transform for kv cache quantization with zero overhead." Proceedings of the AAAI Conference on Artificial Intelligence. Vol. 39. No. 24. 2025.

**Questions:**

How the proposed method performs compared to other KV cache compression methods, including H2O, SnapKV, KIVI and QJL [1] (above)?

Minor issues:
1. in line 160, does ``q ∈ Q_l`` mean a row vector in $Q_l$?
2. in line 180, is the equation should include as Softmax(F(q) F(K_{l,q})^T)? , not K_{k,q}?

---

> ### Author Response · Authors · 2025-11-21
>
> Thank you for the feedback and additional ideas.
>
> ## To answer your questions first:
> 1. How does the proposed method perform compared to other KV cache compression methods?
> QJL is not properly debugged and does not support LLaMA 3.1+ models. H2O is compatible with our method (HuggingFace-based) only in a simulation mode, which makes it impossible to measure real inference runtime. Therefore, we compared our method only with SnapKV and KIVI. However, note that our method is complementary to these approaches, not a competitor. SnapKV works very well, as does KIVI 4-bit, while KIVI 2-bit already drops significantly. But our method keeps most of the performance while improving tokens throughput more that all such models in respect to the base model.
>
> |Model|RULER avg4K|RULER avg8K|tokens/s4k|tokens/s8k|
> |:---|----:|-----:|---:|----:|
> |base Llama3.1 8B Instruct|95.07|93.95|9.79|3.00|
> |DimPO 6.25%|94.29|91.12|9.78|3.12|
> |DimPO 9.38%|93.64|89.48|10.00|3.30|
> |DimPO 12.50%|90.28|85.93|10.42|3.39|
> |KIVI2|94.45|92.64|8.85|3.90|
> |KIVI2 DimPO 6.25%|91.94|86.40|8.99|4.24|
> |KIVI2 DimPO 9.38%|90.52|83.07|9.17|4.44|
> |KIVI2 DimPO 12.50%|81.17|71.05|9.87|5.14|
> |KIVI4|95.05|93.95|8.96|3.93|
> |KIVI4 DimPO 6.25%|94.36|91.35|8.81|3.99|
> |KIVI4 DimPO 9.38%|93.88|89.65|10.82|4.15|
> |KIVI4 DimPO 12.50%|90.33|85.08|11.18|4.34|
> |SnapKV1024|95.07|93.94|9.97|3.98|
> |SnapKV1024 DimPO 6.25%|94.27|91.14|9.97|4.12|
> |SnapKV1024 DimPO 9.38%|93.62|89.47|10.20|4.27|
> |SnapKV1024 DimPO 12.50%|90.27|85.88|10.50|4.47|
> |SnapKV2048|95.07|93.94|9.92|3.96|
> |SnapKV2048 DimPO 6.25%|94.27|91.14|9.94|4.11|
> |SnapKV2048 DimPO 9.38%|93.62|89.47|10.16|4.25|
> |SnapKV2048 DimPO 12.50%|90.27|85.88|10.69|4.46|
>
> 2. For minor issues:
> 2.1 Yes, $q \in Q_l$ denotes a single query vector, i.e., one row of $Q_l \in \mathbb{R}^{N \times d}$ corresponding to a single token.
> 2.2 Thanks for catching this typo, you are right - $\text{Softmax}(F(q), F(K_{l,q})^T)$ is correct.
>
> ## To address your concerns:
> - Additional post-training challenges: you are right that this may be a limitation in some cases. In the paper, we mainly present a new perspective and the potential of the approach, rather than a finished solution that solves all problems. However, for generalization we do not need many training instances: we experimentally found on the validation set that only 10 training instances from the BOOKSUM dataset (completely unrelated to the benchmarks used in Section 4) generalize the projection well for other validation samples; adding more instances does not help. The post-training itself takes some time (though not too much, especially when training all layers in parallel since they are independent), but this is something you do only once, and you can attach the learned checkpoints to the full model’s checkpoints in the same way as downloading pretrained HuggingFace models. Therefore, the post-training time becomes negligible.
>
> - Missing comparison with other methods: we compared with MagicPIG in Table 5, which is not strictly a KV-cache reduction method but sparse attention (yet the effect is similar). We are extending the comparison to SnapKV 1024, SnapKV 2048, and KIVI 2-bit and 4-bit (see table above).
>
> ## Light reframing of the paper’s narrative:
> We found an even better listwise loss function than the described DimPO - ListNet, showing that listwise losses are crucial for Dimensionality Reduction for Attention (and demonstrating that the triplet-based Mongoose approach is not optimal, which is our goal).
> |KLDivergence|d'=64|
> |:----------------|-----------:|
> |Rand|12.8744173|
> |PCA|3.1014843|
> |Triplet|3.2861550|
> |CPO|2.0677807|
> |SimPO|2.8492284|
> |ORPO|2.6025180|
> |RankNet|2.6680312|
> |DimPO|0.7363492|
> |ListNet|0.2656029|
>
> | MSE|d'= 64 |
> |:--------|----------:|
> | Rand    | 0.0792661 |
> | PCA     | 0.0098033 |
> | Triplet | 0.0109971 |
> | CPO     | 0.0084276 |
> | SimPO   | 0.0086739 |
> | ORPO    | 0.0073070 |
> | RankNet | 0.0073032 |
> | DimPO   | 0.0046349 |
> | ListNet | 0.0014229 |
>
> In the end-to-end tasks (for Section 4 in paper), we will show a comparison of different loss functions (listwise as well as pairwise). Here we compare 3 for now, ListNet (listwise) with DimPO (listwise) and ORPO (a well-performing pairwise loss), showing that pairwise methods perform extremely poorly, highlighting the need for listwise losses for dimensionality reduction of attention.
> |Model|RULER avg 4K|tokens/s 4k|
> |:----------|-------:|------------:|
> |SnapKV1024|95.07|9.97|
> |SnapKV1024 DimPO 6.25%|94.27|9.97|
> |SnapKV1024 DimPO 9.38%|93.62|10.20|
> |SnapKV1024 ListNet 6.25%|94.13|9.94|
> |SnapKV1024 ListNet 9.38%|93.65|10.05|
> |SnapKV1024 ORPO 6.25%|65.12|9.95|
> |SnapKV1024 ORPO 9.38%|30.90|10.13|

---

### Author Response · Authors · 2025-12-04
**New Title and TL;DR**

New Title: **Listwise Learning for Dimensionality Reduction of Attention**

TL;DR: We show that listwise preference optimization effectively reduces the dimensionality of attention vectors in LLMs, cutting KV cache memory by 10–15% while preserving over 95% of model performance and remaining compatible with existing KV-cache reduction techniques.


*PDF is going to be resubmitted soon..*

---

### Meta-Review · Area_Chair_KbhZ · 2025-12-28

**Summary:**

(*Disclaimer: given the peculiar review process, some of my choices and reasonings below will be highly subjective, as I tried to imagine how a reviewer would have reacted to a specific response. I understand that any negative choice will be perceived as unfair by the authors, and I apologize in advance for that.*)

(*Second disclaimer: the authors and some reviewers explicitly mention some changes in scores that occurred during the rebuttal. As these were reverted due to the possibility of collusion in light of the security incident, I will tend to disregard this information.*)

The paper proposes a method for KV cache compression, in which keys and queries are mapped to a lower-dimensional space via a contrastive-style loss.

Of the five initial reviews, two reviewers voted strongly for rejection (`1Naz`, `ij1L`), while the other three were clustered on a weak acceptance (`j6MF`, `4Q9G`, `Yw4L`). All reviewers, however, listed similar concerns: poor performance of the original experiments, missing baselines and long-context experiments, and higher training cost as compared to alternative methods.

In the rebuttal, the authors argued that their method is complementary to existing methods, thus the poor performance can be improved by different combinations with other existing techniques. To this end, they also provided several additional experiments, as well as experiments with different loss functions that worked better. They also promised a different PDF with a different title ("*Listwise Learning for Dimensionality Reduction of Attention*"), which was not uploaded, possibly due to the security incident.

Three reviewers never participated in the rebuttal, one reviewer answered but remained negative, and finally the fifth reviewer (`j6MF`) was the only one arguing for (weak) acceptance.

The limited performance of the original submission coupled with the significant changes during rebuttal (including a reframing of the paper and a change in loss functions) highlight that the paper was probably not ready for a review round. Many of the concerns listed by the reviewers were also confirmed by the authors and the (possible) solutions were not discussed during the rebuttal period. Thus, the paper cannot be accepted in this form to ICLR.

**Reviewer Concerns:**

(*I will focus on some key weaknesses identified by multiple reviewers.*)

**Poor performance** (`1Naz`, `j6MF`, `ij1L`): the method incurs a significant drop in performance while providing a limited reduction in the KV cache size. While the new experiments (partially) overcome this, they were a late addition and were not properly discussed or even implemented in the final PDF version.

**Limited evaluation** (`1Naz`, `j6MF`, `4Q9G`, `ij1L`, `Yw4L`): reviewers were concerned about lack of KV cache compression baselines, long-context datasets, and baseline pairwise losses.

**Efficiency** (`1Naz`, `j6MF`, `ij1L`, `Yw4L`): the method incurs a training overhead, which is a non-negligible drawback since it also performs poorly overall. Some reviewers asked for latency comparisons which, however, were available in the original submission.

**Code discrepancies** (`j6MF`): some equations did not match the provided code. While solved, this is another concern pointing to a draft submission that was not properly polished.

**Reviewer Scores:**

`1Naz`: the review was shallow (e.g., no strength) but highlighted most weaknesses of the paper. The reviewer remained unconvinced by the rebuttal.

`j6MF`: the reviewer had similar concerns to the other reviewers while voting for a weak acceptance.

`4Q9G`: another weak acceptance with a focus on limited baselines and datasets. I do not believe the score would have improved since the reviewer never participated in the rebuttal.

`ij1L`: same as `4Q9G`, although the initial score was a strong rejection.

`Yw4L`: same as `4Q9G`.

---

### Decision · Program_Chairs · 2026-01-26

Reject